# New Bounds for Sparse Variational Gaussian Processes

**Michalis K. Titsias** [1]

## Abstract

Sparse variational Gaussian processes (GPs) construct tractable posterior approximations to GP models. At the core of these methods is the assumption that the true posterior distribution over training function values $\mathbf{f}$ and inducing variables $\mathbf{u}$ is approximated by a variational distribution that incorporates the conditional GP prior $p(\mathbf{f}|\mathbf{u})$ in its factorization. While this assumption is considered as fundamental, we show that for model training we can relax it through the use of a more general variational distribution $q(\mathbf{f}|\mathbf{u})$ that depends on $N$ extra parameters, where $N$ is the number of training examples. In GP regression, we can analytically optimize the evidence lower bound over the extra parameters and express a tractable collapsed bound that is tighter than the previous bound. The new bound is also amenable to stochastic optimization and its implementation requires minor modifications to existing sparse GP code. Further, we also describe extensions to non-Gaussian likelihoods. On several datasets we demonstrate that our method can reduce bias when learning the hyperparameters and can lead to better predictive performance.

## 1. Introduction

Gaussian processes (GPs) are nonparametric models for learning functions using Bayesian learning. Thanks to their flexibility and ability to quantify uncertainty, GPs have found many applications in machine learning (Rasmussen & Williams, 2006), spatial modeling (Cressie, 1993), computer experiments (O'Hagan, 1978; Gramacy, 2020), Bayesian optimization (Jones et al., 1998; Garnett, 2023), robotics and control (Deisenroth & Rasmussen, 2011), unsupervised learning (Lawrence, 2005) and others.

Despite the numerous applications, GPs suffer from $\mathcal{O}(N^3)$

[1]Google DeepMind. Correspondence to: Michalis K. Titsias <mtitsias@google.com>.

*Proceedings of the 42$^{nd}$ International Conference on Machine Learning*, Vancouver, Canada. PMLR 267, 2025. Copyright 2025 by the author(s).

time cost and $\mathcal{O}(N^2)$ storage where $N$ is the number of training examples. This has originated a large body of research on scalable or sparse GP methods expanded in several decades; see e.g., Chapter 8 in Rasmussen & Williams (2006) for an early review and Heaton et al. (2018); Liu et al. (2020); Leibfried et al. (2022) for recent treatments. An important class of methods bases an approximation on a small set of $M \ll N$ inducing points (Csato & Opper, 2002; Lawrence et al., 2002; Seeger et al., 2003; Snelson & Ghahramani, 2006; Quiñonero-Candela & Rasmussen, 2005; Banerjee et al., 2008; Finley et al., 2009; Titsias, 2009; Hensman et al., 2013; Bui et al., 2017; Burt et al., 2020) that reduce the time complexity to $\mathcal{O}(NM^2)$ and the storage to $\mathcal{O}(NM)$.

Among inducing point methods, the sparse variational Gaussian process (SVGP), introduced for standard regression (Titsias, 2009), applies variational inference to obtain a posterior approximation and selects hyperparameters and inducing points by maximizing an evidence lower bound. Unlike the prior approximation framework (Quiñonero-Candela & Rasmussen, 2005), SVGP leaves the GP prior unchanged and instead it reduces the cost to $\mathcal{O}(NM^2)$ by imposing a special structure on the variational distribution. This framework has been extended to stochastic gradient optimization (Hensman et al., 2013) and non-Gaussian likelihoods (Chai, 2012; Hensman et al., 2015; Lloyd et al., 2015; Dezfouli & Bonilla, 2015; Sheth et al., 2015). Also, it has been explained as KL minimization between stochastic processes (de G. Matthews et al., 2016).

An important aspect of the SVGP method is that it uses a special form for the variational distribution. It approximates the exact posterior distribution $p(\mathbf{f}, \mathbf{u}|\mathbf{y})$ over the training function values $\mathbf{f}$ and the inducing variables $\mathbf{u}$ (see Section 2 for precise definitions) by a variational distribution of the form $q(\mathbf{f}, \mathbf{u}) = p(\mathbf{f}|\mathbf{u})q(\mathbf{u})$, where $q(\mathbf{u})$ is some optimizable distribution over the inducing variables, while $p(\mathbf{f}|\mathbf{u})$ is the conditional GP prior. This special form of the variational approximation seems to be fundamental, and it has been applied also to more complex GP models, such as those with multiple outputs (Álvarez et al., 2010; Nguyen & Bonilla, 2014; Yousefi et al., 2019), uncertain inputs (Titsias & Lawrence, 2010; Damianou et al., 2016) and multiple layers (Damianou & Lawrence, 2013; Salimbeni & Deisenroth, 2017). However, an open question regarding the SVGP

framework is whether this particular form of variational distribution is really necessary to obtain scalable computations. The answer we give in this paper is that "it is not", since at least for training a GP model it can be relaxed.

To this end, we derive new variational bounds for training sparse GP regression models by replacing $p(\mathbf{f}|\mathbf{u})$ in the variational distribution with a more general conditional distribution $q(\mathbf{f}|\mathbf{u})$. This $q(\mathbf{f}|\mathbf{u})$ depends on $N$ additional parameters (on top of the parameters of $p(\mathbf{f}|\mathbf{u})$), i.e., as many as the training examples, and it is constructed to enable better covariance approximation of the underlying true factor $p(\mathbf{f}|\mathbf{u},\mathbf{y})$. We show how to analytically optimize over the $N$ parameters and obtain a better posterior approximation together with a tighter collapsed evidence lower bound. The new bound is also amenable to stochastic gradient optimization, and its simple form suggests that it can be implemented with minor modifications to existing sparse GP code. We also describe extensions of the method to non-Gaussian likelihoods. Furthermore, we point out the concurrent work of Bui et al. (2025) who derived similar sparse GP approximations and variational training objectives by using the same form for the $q(\mathbf{f}|\mathbf{u})$ distribution.

The remainder of the paper is as follows. Section 2 provides an overview of GPs and the variational approach to sparse GPs using inducing points. Section 3 derives the new evidence lower bounds for training. Section 4 discusses connections with previous works. Section 5 presents experiments using several datasets showing that the new bounds can reduce underfitting bias and can lead to better predictive performance. Section 6 concludes with a discussion and suggestions for future work.

## 2. Background

A GP is a distribution over functions specified by a mean function $m(\boldsymbol{x})$ and a covariance or kernel function $k(\boldsymbol{x},\boldsymbol{x}')$, where the kernel function is parametrized by $\theta$. By assuming that $m(\boldsymbol{x})=0$ we denote a GP draw as

$$f(\boldsymbol{x}) \sim \mathcal{GP}(0, k(\boldsymbol{x},\boldsymbol{x}')).$$

For a finite set of inputs $\mathbf{X}=\{\boldsymbol{x}_n\}_{n=1}^N$ the distribution over the function values $\mathbf{f}=\{f_n\}_{n=1}^N$ (stored as $N \times 1$ vector with $f_n := f(\boldsymbol{x}_n)$) is the multivariate Gaussian $p(\mathbf{f})=\mathcal{N}(\mathbf{f}|\mathbf{0},\mathbf{K_{ff}})$ where the $N \times N$ covariance matrix $\mathbf{K_{ff}}$ has entries $[\mathbf{K_{ff}}]_{ij}=k(\boldsymbol{x}_i,\boldsymbol{x}_j)$.

We consider standard GP regression where we are given a set of training inputs $\mathbf{X}$ and corresponding noisy outputs $\mathbf{y}=\{y_n\}_{n=1}^N$ where $y_n \in \mathbb{R}$. Conditionally on the latent values $\mathbf{f}$, these outputs follow a factorized Gaussian likelihood, $p(\mathbf{y}|\mathbf{f})=\prod_{n=1}^N \mathcal{N}(y_n|f_n,\sigma^2)=\mathcal{N}(\mathbf{y}|\mathbf{f},\sigma^2 I)$. The joint distribution over outputs $\mathbf{y}$ and latent values $\mathbf{f}$ is

$$p(\mathbf{y}|\mathbf{f})p(\mathbf{f}) = \mathcal{N}(\mathbf{y}|\mathbf{f},\sigma^2 I)\mathcal{N}(\mathbf{f}|\mathbf{0},\mathbf{K_{ff}}). \quad (1)$$

To learn the hyperparameters $(\theta,\sigma^2)$ we can maximize the log marginal likelihood which is analytically available,

$$\log p(\mathbf{y}) = \int p(\mathbf{y}|\mathbf{f})p(\mathbf{f})d\mathbf{f} = \log \mathcal{N}(\mathbf{y}|\mathbf{0},\mathbf{K_{ff}}+\sigma^2\mathbf{I}). \quad (2)$$

After training we can perform predictions at test inputs $\mathbf{X}_*$ by first computing the posterior over the corresponding test function values $\mathbf{f}_*$:

$$p(\mathbf{f}_*|\mathbf{y}) = \int p(\mathbf{f}_*|\mathbf{f})p(\mathbf{f}|\mathbf{y})d\mathbf{f} = \quad (3)$$

$$\mathcal{N}(\mathbf{f}_*|\mathbf{K_{f_*f}}(\mathbf{K_{ff}}+\sigma^2\mathbf{I})^{-1}\mathbf{f}, \mathbf{K_{f_*f_*}}-\mathbf{K_{f_*f}}(\mathbf{K_{ff}}+\sigma^2\mathbf{I})^{-1}\mathbf{K_{ff_*}})$$

and then writing the predictive density as $p(\mathbf{y}_*|\mathbf{y}) = \int \mathcal{N}(\mathbf{y}_*|\mathbf{f}_*,\sigma^2\mathbf{I})p(\mathbf{f}_*|\mathbf{y})d\mathbf{f}_*$, which is the same as the above Gaussian but with $\sigma^2\mathbf{I}$ added to the covariance.

While the log marginal likelihood and predictive density have closed-form expressions, they require the inversion of $\mathbf{K_{ff}}+\sigma^2\mathbf{I}$ which costs $\mathcal{O}(N^3)$ and it is prohibitive for large datasets. Next we review methods using inducing points and particularly the variational approach (Titsias, 2009) that our method in Section 3 improves upon.

### 2.1. Sparse Variational Gaussian Process (SVGP)

The idea of inducing points is to base a GP approximation on a smaller set of $M \ll N$ function values; see e.g., Csato & Opper (2002); Seeger et al. (2003); Snelson & Ghahramani (2006); Quiñonero-Candela & Rasmussen (2005). Snelson & Ghahramani (2006) introduced pseudo inputs by instantiating extra GP function values $\mathbf{u}=\{f(\boldsymbol{z}_m)\}_{m=1}^M$ evaluated at locations $\mathbf{Z}=\{\boldsymbol{z}_m\}_{m=1}^M$ that can be optimized freely with gradient-based methods. However, the GP prior modification procedure (Quiñonero-Candela & Rasmussen, 2005; Snelson & Ghahramani, 2006) does not result in a rigorous approximation to the GP model. An alternative variational inference method (Titsias, 2009), next referred to as SVGP[1], does not modify the GP prior but instead it augments the model with extra function values $\mathbf{u}$:

$$p(\mathbf{y},\mathbf{f},\mathbf{u})=p(\mathbf{y}|\mathbf{f})p(\mathbf{f}|\mathbf{u})p(\mathbf{u}) \quad \text{augmented joint} \quad (4)$$

$$p(\mathbf{f}|\mathbf{u})=\mathcal{N}(\mathbf{f}|\mathbf{K_{ff}}\mathbf{K_{uu}^{-1}}\mathbf{u}, \mathbf{K_{ff}}-\mathbf{K_{fu}}\mathbf{K_{uu}^{-1}}\mathbf{K_{uf}}) \quad \text{cond. GP} \quad (5)$$

$$p(\mathbf{u})=\mathcal{N}(\mathbf{u}|\mathbf{0},\mathbf{K_{uu}}) \quad \text{inducing GP prior} \quad (6)$$

where $\mathbf{K_{uu}}$ is the $M \times M$ covariance matrix on the inducing inputs $\mathbf{Z}$, $\mathbf{K_{fu}}$ is the $N \times M$ cross covariance between points in $\mathbf{X}$ and $\mathbf{Z}$, while $\mathbf{K_{uf}}=\mathbf{K_{fu}^\top}$. SVGP approximates the exact posterior $p(\mathbf{f},\mathbf{u}|\mathbf{y})$ by a variational distribution $q(\mathbf{f},\mathbf{u})$ through the minimization of $\mathrm{KL}[q(\mathbf{f},\mathbf{u})||p(\mathbf{f},\mathbf{u}|\mathbf{y})]$.

[1]Another common name for this method is Variational Free Energy (VFE); see Bui et al. (2017); Bauer et al. (2016); Liu et al. (2020).

A critical assumption is the following choice for $q$:

$$q(\mathbf{f}, \mathbf{u}) = p(\mathbf{f}|\mathbf{u})q(\mathbf{u}), \quad (7)$$

where $q(\mathbf{u})$ is an optimizable $M$-dimensional variational distribution, while $p(\mathbf{f}|\mathbf{u})$ is the same conditional GP prior from Equation (5) that appears in the joint in (4). The KL minimization is expressed as the maximization of an evidence lower bound (ELBO) on the log marginal likelihood,

$$\log p(\mathbf{y}) \geq \int p(\mathbf{f}|\mathbf{u})q(\mathbf{u}) \log \frac{p(\mathbf{y}|\mathbf{f})p(\mathbf{f}|\mathbf{u})p(\mathbf{u})}{p(\mathbf{f}|\mathbf{u})q(\mathbf{u})} d\mathbf{f}d\mathbf{u}$$

$$= \int q(\mathbf{u}) \log \frac{\exp\{\int p(\mathbf{f}|\mathbf{u}) \log p(\mathbf{y}|\mathbf{f})d\mathbf{f}\}p(\mathbf{u})}{q(\mathbf{u})} d\mathbf{u}.$$

If we optimize over $q(\mathbf{u})$ and obtain the optimal choice $q^*(\mathbf{u}) \propto \exp\left\{ \int p(\mathbf{f}|\mathbf{u}) \log p(\mathbf{y}|\mathbf{f})d\mathbf{f} \right\} p(\mathbf{u})$, then we can substitute this $q^*(\mathbf{u})$ in the last line above and express the so called collapsed bound, having the general form

$$\log p(\mathbf{y}) \geq \mathcal{F} = \log \int \exp\left\{ \int p(\mathbf{f}|\mathbf{u}) \log p(\mathbf{y}|\mathbf{f})d\mathbf{f} \right\} p(\mathbf{u})d\mathbf{u}$$

which for the standard GP regression model takes the form

$$\mathcal{F} = \underbrace{\log \mathcal{N}(\mathbf{y}|\mathbf{0}, \mathbf{Q}_{\mathbf{ff}} + \sigma^2\mathbf{I})}_{\text{DTC log lik}} - \underbrace{\frac{1}{2\sigma^2}\text{tr}(\mathbf{K}_{\mathbf{ff}} - \mathbf{Q}_{\mathbf{ff}})}_{\text{trace term}}, \quad (8)$$

where $\mathbf{Q}_{\mathbf{ff}} = \mathbf{K}_{\mathbf{fu}}\mathbf{K}_{\mathbf{uu}}^{-1}\mathbf{K}_{\mathbf{uf}}$ is the M-rank Nyström matrix. The first term in the bound is the deterministic training conditional (DTC) log likelihood (Seeger et al., 2003; Quiñonero-Candela & Rasmussen, 2005) while the second is a regularization term which, since $\text{tr}(\mathbf{K}_{\mathbf{ff}} - \mathbf{Q}_{\mathbf{ff}}) \geq 0$, promotes $\mathbf{Q}_{\mathbf{ff}}$ to stay close to $\mathbf{K}_{\mathbf{ff}}$. The inducing points $\mathbf{Z}$ can be learned as variational parameters by maximizing the bound jointly with the hyperparameters $(\theta, \sigma^2)$, which requires $\mathcal{O}(NM^2)$ operations per optimization step. Hensman et al. (2013) further reduced the operations to $\mathcal{O}(M^3)$ per optimization step by applying stochastic minibatch training for maximizing the uncollapsed version of the bound; see Section 3.2.

To obtain the form of the GP posterior over any test function values $\mathbf{f}_*$ we can first write the exact form

$$p(\mathbf{f}_*|\mathbf{y}) = \int p(\mathbf{f}_*|\mathbf{f}, \mathbf{u})p(\mathbf{f}, \mathbf{u}|\mathbf{y})d\mathbf{f}d\mathbf{u}, \quad (9)$$

where $p(\mathbf{f}_*|\mathbf{f}, \mathbf{u})$ is the conditional GP of $\mathbf{f}_*$ given training function values $\mathbf{f}$ and inducing values $\mathbf{u}$ while $p(\mathbf{f}, \mathbf{u}|\mathbf{y})$ is the posterior over $(\mathbf{f}, \mathbf{u})$ written also as

$$p(\mathbf{f}, \mathbf{u}|\mathbf{y}) = p(\mathbf{f}|\mathbf{u}, \mathbf{y})p(\mathbf{u}|\mathbf{y}). \quad (10)$$

The SVGP method approximates $p(\mathbf{f}, \mathbf{u}|\mathbf{y})$ by $q(\mathbf{f}, \mathbf{u})$ and therefore by plugging in this $q$ into (9) we obtain

$$q(\mathbf{f}_*|\mathbf{y}) = \int p(\mathbf{f}_*|\mathbf{f}, \mathbf{u})p(\mathbf{f}|\mathbf{u})q(\mathbf{u})d\mathbf{f}d\mathbf{u} = \int p(\mathbf{f}_*|\mathbf{u})q(\mathbf{u})d\mathbf{u}, \quad (11)$$

where $p(\mathbf{f}_*|\mathbf{u}) = \int p(\mathbf{f}_*|\mathbf{f}, \mathbf{u})p(\mathbf{f}|\mathbf{u})d\mathbf{f}$ comes from the GP consistency. For completeness, in Appendix A we include further details about SVGP such as a derivation of the collapsed bound and the Gaussian form of the optimal $q^*(\mathbf{u})$.

We conclude this review of SVGP for regression with a couple of remarks that will be useful next.

*Remark* 2.1. The approximation becomes exact when $\mathbf{K}_{\mathbf{ff}} = \mathbf{Q}_{\mathbf{ff}}$ and the collapsed bound matches the log marginal likelihood in (2). However, to obtain good approximations we may need sufficiently large number of inducing points (Burt et al., 2020). Otherwise the bound will cause underfitting. For instance, as studied by Bauer et al. (2016) and Titsias (2009) the SVGP bound tends to overestimate the noise variance $\sigma^2$.

*Remark* 2.2. SVGP approximates $p(\mathbf{f}|\mathbf{u}, \mathbf{y})$ in the exact posterior in (10) by the conditional GP $p(\mathbf{f}|\mathbf{u})$, in the variational posterior in (7), while $q(\mathbf{u})$ is treated optimally by KL minimization. If $p(\mathbf{f}|\mathbf{u}, \mathbf{y}) = p(\mathbf{f}|\mathbf{u})$ then $\text{KL}[q(\mathbf{f}, \mathbf{u})||p(\mathbf{f}, \mathbf{u}|\mathbf{y})] = 0$ and the approximation becomes exact, meaning $q(\mathbf{f}_*|\mathbf{y}) = p(\mathbf{f}_*|\mathbf{y})$ for any $\mathbf{f}_* = f_*(\mathbf{X}_*)$.

## 3. Proposed Method: Tighter Bounds

Remark 2.1 suggests that it would be useful to tighten the collapsed bound in order to reduce underfitting bias and match better exact GP training. Remark 2.2 suggests that one way to tighten the bound is to replace $p(\mathbf{f}|\mathbf{u})$, in the variational approximation in (7), with another distribution that can better approximate $p(\mathbf{f}|\mathbf{u}, \mathbf{y})$. Next we develop a method that does this while keeping the cost unchanged.

Let us write the exact form of $p(\mathbf{f}|\mathbf{u}, \mathbf{y})$. By noting that this quantity is the exact posterior over $\mathbf{f}$ in a GP regression model with joint $p(\mathbf{y}|\mathbf{f})p(\mathbf{f}|\mathbf{u})$ we conclude that this is

$$p(\mathbf{f}|\mathbf{u}, \mathbf{y}) = \mathcal{N}\left( \mathbf{f}|\mathbf{m}(\mathbf{y}, \mathbf{u}), (\widetilde{\mathbf{K}}_{\mathbf{ff}}^{-1} + \frac{1}{\sigma^2}\mathbf{I})^{-1} \right),$$

where $\mathbf{m}(\mathbf{y}, \mathbf{u}) = \mathbb{E}[\mathbf{f}|\mathbf{u}] + \widetilde{\mathbf{K}}_{\mathbf{ff}}(\widetilde{\mathbf{K}}_{\mathbf{ff}} + \sigma^2\mathbf{I})^{-1}(\mathbf{y} - \mathbb{E}[\mathbf{f}|\mathbf{u}])$ with $\mathbb{E}[\mathbf{f}|\mathbf{u}] = \mathbf{K}_{\mathbf{fu}}\mathbf{K}_{\mathbf{uu}}^{-1}\mathbf{u}$ and $\widetilde{\mathbf{K}}_{\mathbf{ff}} = \mathbf{K}_{\mathbf{ff}} - \mathbf{Q}_{\mathbf{ff}}$. Note that under this notation, $p(\mathbf{f}|\mathbf{u}) = \mathcal{N}(\mathbf{f}|\mathbb{E}[\mathbf{f}|\mathbf{u}], \widetilde{\mathbf{K}}_{\mathbf{ff}})$. We will construct a new $q(\mathbf{f}|\mathbf{u})$ that keeps the same mean $\mathbb{E}[\mathbf{f}|\mathbf{u}]$ as $p(\mathbf{f}|\mathbf{u})$ but it replaces $\widetilde{\mathbf{K}}_{\mathbf{ff}}$ with a closer approximation to the covariance $(\widetilde{\mathbf{K}}_{\mathbf{ff}}^{-1} + \frac{1}{\sigma^2}\mathbf{I})^{-1}$ of $p(\mathbf{f}|\mathbf{u}, \mathbf{y})$. We first write this matrix as

$$(\widetilde{\mathbf{K}}_{\mathbf{ff}}^{-1} + \frac{1}{\sigma^2}\mathbf{I})^{-1} = \widetilde{\mathbf{K}}_{\mathbf{ff}}^{\frac{1}{2}}(\mathbf{I} + \frac{1}{\sigma^2}\widetilde{\mathbf{K}}_{\mathbf{ff}})^{-1}\widetilde{\mathbf{K}}_{\mathbf{ff}}^{\frac{1}{2}}. \quad (12)$$

Then we approximate the inverse $(\mathbf{I} + \frac{1}{\sigma^2}\widetilde{\mathbf{K}}_{\mathbf{ff}})^{-1}$ by a diagonal matrix $\mathbf{V} = \text{diag}(v_1, \ldots, v_N)$ of $N$ variational parameters $v_i > 0$. In other words, in the initial $q(\mathbf{f}, \mathbf{u}) = p(\mathbf{f}|\mathbf{u})q(\mathbf{u})$ we will replace $p(\mathbf{f}|\mathbf{u})$ by

$$q(\mathbf{f}|\mathbf{u}) = \mathcal{N}(\mathbf{f}|\mathbf{K}_{\mathbf{fu}}\mathbf{K}_{\mathbf{uu}}^{-1}\mathbf{u}, (\mathbf{K}_{\mathbf{ff}} - \mathbf{Q}_{\mathbf{ff}})^{\frac{1}{2}}\mathbf{V}(\mathbf{K}_{\mathbf{ff}} - \mathbf{Q}_{\mathbf{ff}})^{\frac{1}{2}}). \quad (13)$$

The ELBO now is written as

$$\int q(\mathbf{f}|\mathbf{u})q(\mathbf{u})\log\frac{p(\mathbf{y}|\mathbf{f})p(\mathbf{f}|\mathbf{u})p(\mathbf{u})}{q(\mathbf{f}|\mathbf{u})q(\mathbf{u})}d\mathbf{f}d\mathbf{u} =$$

$$\int q(\mathbf{u})\left\{\log\frac{e^{\mathbb{E}_{q(\mathbf{f}|\mathbf{u})}[\log p(\mathbf{y}|\mathbf{f})]}p(\mathbf{u})}{q(\mathbf{u})}-\mathrm{KL}[q(\mathbf{f}|\mathbf{u})||p(\mathbf{f}|\mathbf{u})]\right\}d\mathbf{u}$$

and the challenge is to see whether $\mathrm{KL}[q(\mathbf{f}|\mathbf{u})||p(\mathbf{f}|\mathbf{u})]$ and $\mathbb{E}_{q(\mathbf{f}|\mathbf{u})}[\log p(\mathbf{y}|\mathbf{f})]$ are computable in $\mathcal{O}(NM^2)$ time. We have the following results (proofs are in Appendix B).

**Lemma 3.1.** $\mathrm{KL}[q(\mathbf{f}|\mathbf{u})||p(\mathbf{f}|\mathbf{u})] = \frac{1}{2}\sum_{i=1}^{N}(v_i - \log v_i - 1).$

**Lemma 3.2.** *Let us denote the diagonal elements of $\mathbf{K_{ff}} - \mathbf{Q_{ff}}$ as $k_{ii} - q_{ii}$ for $i = 1,\ldots,N$. Then*

$$\mathbb{E}_{q(\mathbf{f}|\mathbf{u})}[\log p(\mathbf{y}|\mathbf{f})]$$

$$= \log\mathcal{N}(\mathbf{y}|\mathbf{K_{fu}}\mathbf{K_{uu}}^{-1}\mathbf{u}, \sigma^2\mathbf{I}) - \frac{1}{2\sigma^2}\sum_{i=1}^{N}v_i(k_{ii}-q_{ii}). \quad (14)$$

By combining the two lemmas the full bound is written as

$$\int q(\mathbf{u})\log\frac{\mathcal{N}(\mathbf{y}|\mathbf{K_{fu}}\mathbf{K_{uu}}^{-1}\mathbf{u}, \sigma^2\mathbf{I})p(\mathbf{u})}{q(\mathbf{u})}d\mathbf{u}$$

$$-\frac{1}{2}\sum_{i=1}^{N}\left\{v_i\left(1 + \frac{k_{ii}-q_{ii}}{\sigma^2}\right) - \log v_i - 1\right\}. \quad (15)$$

**Proposition 3.3.** *Maximizing the bound in (15) with respect to $q(\mathbf{u})$ and each $v_i$ results in the optimal settings $q^*(\mathbf{u}) \propto \mathcal{N}(\mathbf{y}|\mathbf{K_{fu}}\mathbf{K_{uu}}^{-1}\mathbf{u}, \sigma^2\mathbf{I})p(\mathbf{u})$ and $v_i^* = \left(1 + \frac{k_{ii}-q_{ii}}{\sigma^2}\right)^{-1}$. By substituting these values back to (15) we obtain*

$$\mathcal{F}_{new} = \log\mathcal{N}(\mathbf{y}|\mathbf{0}, \mathbf{Q_{ff}} + \sigma^2\mathbf{I}) - \frac{1}{2}\sum_{i=1}^{N}\log\left(1 + \frac{k_{ii}-q_{ii}}{\sigma^2}\right). \quad (16)$$

The first term is the DTC log likelihood as in the original bound in (8), but the regularization term makes the bound tighter, i.e., $\log p(\mathbf{y}) \geq \mathcal{F}_{new} \geq \mathcal{F}$, due to the inequality $\log(a+1) \leq a$. Also since $\log(a+1) < a$ for all $a > 0$, if $\mathbf{K_{ff}} \neq \mathbf{Q_{ff}}$ (so there is at least one $k_{ii} - q_{ii} > 0$), then $\mathcal{F}_{new} > \mathcal{F}$. This means that $\mathcal{F}_{new}$ is strictly better than $\mathcal{F}$ unless both bounds match exactly the log marginal likelihood.

Clearly, $\mathcal{F}_{new}$ has $\mathcal{O}(NM^2)$ cost and its implementation requires a minor modification to the initial bound. The optimal $q^*(\mathbf{u})$ is the same as in the initial SVGP method, while an interpretation of the optimal $v_i^*$ values is the following.

*Remark* 3.4. The diagonal matrix $\mathbf{V}^*$ (with the optimal $v_i^*$ values in its diagonal) is the inverse obtained after zeroing out the off-diagonal elements of $\mathbf{I} + \frac{1}{\sigma^2}(\mathbf{K_{ff}} - \mathbf{Q_{ff}})$, i.e., $\mathbf{V}^* = \mathrm{diag}[\mathbf{I} + \frac{1}{\sigma^2}(\mathbf{K_{ff}} - \mathbf{Q_{ff}})]^{-1}$ which approximates $(\mathbf{I} +$

$\frac{1}{\sigma^2}(\mathbf{K_{ff}} - \mathbf{Q_{ff}}))^{-1}$ in Equation (12). Also note that in the ordering of positive definite matrices it holds $\mathbf{V}^* \leq \mathbf{I}$, from which it follows that $q(\mathbf{f}|\mathbf{u})$ has smaller covariance than $p(\mathbf{f}|\mathbf{u})$ and more accurately approximates the covariance of $p(\mathbf{f}|\mathbf{u}, \mathbf{y})$. The latter as implied by Equation (12), has also smaller covariance than $p(\mathbf{f}|\mathbf{u})$.

### 3.1. Predictions

To perform predictions we will be using the same predictive posterior from Equation (11), i.e., $q(\mathbf{f}_*|\mathbf{y}) = \int p(\mathbf{f}_*|\mathbf{u})q(\mathbf{u})d\mathbf{u}$, where the optimal $q^*(\mathbf{u})$ (see Appendix A) is exactly the same as in the standard SVGP method. The alternative expression (and strictly speaking more appropriate since our variational approximation is $q(\mathbf{f}|\mathbf{u})q(\mathbf{u})$) is given by

$$q_{high\_cost}(\mathbf{f}_*|\mathbf{y}) = \int p(\mathbf{f}_*|\mathbf{f}, \mathbf{u})q(\mathbf{f}|\mathbf{u})q(\mathbf{u})d\mathbf{f}d\mathbf{u}. \quad (17)$$

But this is expensive since it has cost $\mathcal{O}(N^3)$. The reason is that $\int p(\mathbf{f}_*|\mathbf{f}, \mathbf{u})q(\mathbf{f}|\mathbf{u})d\mathbf{f}$ does not simplify anymore since $q(\mathbf{f}|\mathbf{u})$ is not the conditional GP, which means that $p(\mathbf{f}_*|\mathbf{f}, \mathbf{u})$ and $q(\mathbf{f}|\mathbf{u})$ are not consistent under the GP prior. Nevertheless, $q(\mathbf{f}_*|\mathbf{y})$ and $q_{high\_cost}(\mathbf{f}_*|\mathbf{y})$ have exactly the same mean, since $q(\mathbf{f}|\mathbf{u})$ and $p(\mathbf{f}|\mathbf{u})$ have the same mean, but the tractable $q$ will give higher variances than $q_{high\_cost}$.

### 3.2. Stochastic Minibatch Training

The initial SVGP method (Titsias, 2009) does the training in a batch mode where all data are used in each optimization step. Stochastic optimization using minibatches was proposed by Hensman et al. (2013). Here, we apply our new approximation to this stochastic method.

We start from Equation (15), and substitute only the optimal values for each $v_i$ without using the optimal setting for $q(\mathbf{u})$. This results in the uncollapsed bound

$$\sum_{i=1}^{N}\left\{\mathbb{E}_{q(\mathbf{u})}[\log\mathcal{N}(y_i|\mathbf{k}_{f_i\mathbf{u}}\mathbf{K_{uu}}^{-1}\mathbf{u}, \sigma^2)]\right.$$

$$\left. -\frac{1}{2}\log\left(1 + \frac{k_{ii}-q_{ii}}{\sigma^2}\right)\right\} - \mathrm{KL}[q(\mathbf{u})||p(\mathbf{u})], \quad (18)$$

where $\mathbf{k}_{f_i\mathbf{u}}$ is the $1 \times M$ vector of all kernel values between the training input $\mathbf{x}_i$ and the inducing inputs $\mathbf{Z}$, while the expectation under $q(\mathbf{u})$ in the first line is analytic; see Hensman et al. (2013). The above bound is strictly better than the previous uncollapsed bound in Hensman et al. (2013), since $-\frac{1}{2\sigma^2}(k_{ii} - q_{ii}) \leq -\frac{1}{2}\log\left(1 + \frac{k_{ii}-q_{ii}}{\sigma^2}\right)$. Based on the above we can apply stochastic gradient methods to optimize $q(\mathbf{u})$ and the hyperparameters by subsampling data minibatches to deal with the sum over the $N$ training points,

i.e., at each iteration we use the stochastic ELBO:

$$\frac{N}{|\mathcal{B}|} \sum_{i \in \mathcal{B}} \left\{ \mathbb{E}_{q(\mathbf{u})}[\log \mathcal{N}(y_i | \mathbf{k}_{f_i \mathbf{u}} \mathbf{K}_{\mathbf{uu}}^{-1} \mathbf{u}, \sigma^2)] \right.$$
$$\left. - \frac{1}{2} \log \left( 1 + \frac{k_{ii} - q_{ii}}{\sigma^2} \right) \right\} - \mathrm{KL}[q(\mathbf{u})||p(\mathbf{u})], \quad (19)$$

where $\mathcal{B}$ denotes a minibatch.

The most common parametrization of $q(\mathbf{u})$ is $q(\mathbf{u}) = \mathcal{N}(\mathbf{u}|\mathbf{m}, \mathbf{S})$ where the mean vector $\mathbf{m}$ and covariance matrix $\mathbf{S}$ are variational parameters. Another popular parametrization, for instance used as the default in GPflow (de G. Matthews et al., 2017), is the whitened parametrization that we consider in our experiments. For any choice of $q(\mathbf{u})$, the new bound is always tighter than its corresponding previous uncollapsed bound and requires minor modifications to existing implementations, i.e., to replace the previous term $-\frac{1}{2\sigma^2}(k_{ii} - q_{ii})$ with $-\frac{1}{2} \log \left( 1 + \frac{k_{ii} - q_{ii}}{\sigma^2} \right)$.

### 3.3. Non-Gaussian Likelihoods

Consider a factorized likelihood $p(\mathbf{y}|\mathbf{f}) = \prod_{i=1}^{N} p(y_i|f_i)$ where $p(y_i|f_i)$ is non-Gaussian, e.g., Bernoulli for binary outputs or Poisson for counts. In this non-conjugate setting the sparse variational GP approximation imposes the same form for the variational distribution, i.e., $q(\mathbf{f}, \mathbf{u}) = p(\mathbf{f}|\mathbf{u})q(\mathbf{u})$ where $p(\mathbf{f}|\mathbf{u})$ is the conditional GP prior. As shown in several works (Chai, 2012; Hensman et al., 2015; Lloyd et al., 2015; Dezfouli & Bonilla, 2015; Sheth et al., 2015), this leads to the bound

$$\sum_{i=1}^{N} \mathbb{E}_{q(f_i)}[\log p(y_i|f_i)] - \mathrm{KL}[q(\mathbf{u})||p(\mathbf{u})], \quad (20)$$

where $q(f_i) = \int p(\mathbf{f}|\mathbf{u})q(\mathbf{u})d\mathbf{f}_{-i}d\mathbf{u}$ is the marginal over $f_i := f(\boldsymbol{x}_i)$ with respect to the approximate posterior $q(\mathbf{f}, \mathbf{u})$. Given that $q(\mathbf{u})$ is Gaussian with mean $\mathbf{m}$ and covariance $\mathbf{S}$, $q(f_i)$ can be computed fast in $\mathcal{O}(M^2)$ time (after precomputing the Cholesky factorization of $\mathbf{K}_{\mathbf{uu}}$) as follows

$$q(f_i) = \mathcal{N}(f_i | \mathbf{k}_{f_i \mathbf{u}} \mathbf{K}_{\mathbf{uu}}^{-1} \mathbf{m}, k_{ii} - q_{ii} + \mathbf{k}_{f_i \mathbf{u}} \mathbf{K}_{\mathbf{uu}}^{-1} \mathbf{S} \mathbf{K}_{\mathbf{uu}}^{-1} \mathbf{k}_{\mathbf{u} f_i}).$$
$$(21)$$

For the discussion next it is useful to observe that the efficiency when computing $q(f_i)$ comes from $p(\mathbf{f}|\mathbf{u})$ being a conditional GP prior, so expressing $p(f_i|\mathbf{u})$ is trivial.

Suppose now that we wish to impose the more structured variational approximation $q(\mathbf{f}, \mathbf{u}) = q(\mathbf{f}|\mathbf{u})q(\mathbf{u})$ where $q(\mathbf{u}) = \mathcal{N}(\mathbf{u}|\mathbf{m}, \mathbf{S})$ and $q(\mathbf{f}|\mathbf{u})$ is given by Equation (13). The bound can be written as

$$\sum_{i=1}^{N} \mathbb{E}_{q(f_i)}[\log p(y_i|f_i)] - \frac{1}{2} \sum_{i=1}^{N} (v_i - \log v_i - 1)$$
$$- \mathrm{KL}[q(\mathbf{u})||p(\mathbf{u})], \quad (22)$$

where we used the fact that $\mathrm{KL}[q(\mathbf{f}|\mathbf{u})||p(\mathbf{f}|\mathbf{u})]$ is obtained from Lemma 3.1. The above bound is not computationally efficient since the marginal $q(f_i) = \int q(\mathbf{f}|\mathbf{u})q(\mathbf{u})d\mathbf{f}_{-i}d\mathbf{u}$ has $\mathcal{O}(N^3)$ cost. This is because the marginalization $q(f_i|\mathbf{u}) = \int q(\mathbf{f}|\mathbf{u})d\mathbf{f}_{-i}$ cannot be trivially expressed, due to the complex structure of the covariance $(\mathbf{K}_{\mathbf{ff}} - \mathbf{Q}_{\mathbf{ff}})^{\frac{1}{2}} \mathbf{V} (\mathbf{K}_{\mathbf{ff}} - \mathbf{Q}_{\mathbf{ff}})^{\frac{1}{2}}$ in $q(\mathbf{f}|\mathbf{u})$. To overcome this, we will use a simplified version of $q(\mathbf{f}|\mathbf{u})$, in which we choose a spherical $\mathbf{V} = v\mathbf{I}$ with $v > 0$. Then, things become tractable.

**Proposition 3.5.** *Let* $q(\mathbf{f}|\mathbf{u}) = \mathcal{N}(\mathbf{f}|\mathbf{K}_{\mathbf{fu}}\mathbf{K}_{\mathbf{uu}}^{-1}\mathbf{u}, v(\mathbf{K}_{\mathbf{ff}} - \mathbf{Q}_{\mathbf{ff}}))$ *for* $v > 0$. *Then* (22) *is computed in* $\mathcal{O}(NM^2)$ *time as*

$$\sum_{i=1}^{N} \mathbb{E}_{q(f_i)}[\log p(y_i|f_i)] - \frac{N}{2}(v - \log v - 1)$$
$$- \mathrm{KL}[q(\mathbf{u})||p(\mathbf{u})], \quad (23)$$

*where the marginal is* $q(f_i) = \mathcal{N}(f_i | \mathbf{k}_{f_i \mathbf{u}} \mathbf{K}_{\mathbf{uu}}^{-1} \mathbf{m}, v(k_{ii} - q_{ii}) + \mathbf{k}_{f_i \mathbf{u}} \mathbf{K}_{\mathbf{uu}}^{-1} \mathbf{S} \mathbf{K}_{\mathbf{uu}}^{-1} \mathbf{k}_{\mathbf{u} f_i})$.

The parameter $v$ multiplies the term $k_{ii} - q_{ii}$ inside the variance of $q(f_i)$, and it also appears in the regularization term $-\frac{N}{2}(v - \log v - 1)$. If $v = 1$ the bound in (23) reduces to (20), while by optimizing over $v$ it can become a tighter bound. The optimization of $v$ is done jointly with the remaining parameters $\mathbf{m}, \mathbf{S}, \mathbf{Z}, \theta$ using gradient-based methods. Stochastic gradients can also be used by subsampling minibatches to deal with the sum $\sum_{i=1}^{N} \mathbb{E}_{q(f_i)}[\log p(y_i|f_i)]$ and reduce the complexity to $\mathcal{O}(M^3)$.

The above framework can be extended to non-conjugate models having multiple functions, such as multi-class GP classification, by introducing a separate $v$ parameter per GP function. In our experiments, we consider only single-function non-conjugate GP models and we leave the experimentation with more complex models for future work.

## 4. Related Work

Several recent works on sparse GPs focus on constructing efficient inducing points, such as works that place inducing points on a grid (Wilson & Nickisch, 2015; Evans & Nair, 2018; Gardner et al., 2018), construct inter-domain Fourier features (Lázaro-Gredilla & Figueiras-Vidal, 2009; Hensman et al., 2018), provide Bayesian treatments to inducing inputs (Rossi et al., 2021) or use nearest neighbor sparsity structures (Tran et al., 2021; Wu et al., 2022). There exist also algorithms that allow to increase the number of inducing points using the decoupled method (Cheng & Boots, 2017; Havasi et al., 2018) and the related orthogonally decoupled approaches (Salimbeni et al., 2018; Shi et al., 2020; Sun et al., 2021; Tiao et al., 2023). Our contribution is orthogonal to these previous methods since we relax the conditional GP prior assumption in the posterior variational

approximation. This means that our method could be used to improve previous variational sparse GP approaches, as the ones mentioned above as well as earlier schemes that select inducing points from the training inputs (Cao et al., 2013; Chai, 2012; Schreiter et al., 2016).

Zhu et al. (2023) proposed inducing points GP approximations that change the conditional GP $p(\mathbf{f}|\mathbf{u})$ in the variational approximation to a modified conditional GP that uses different kernel hyperparameters in its mean vector. Note that our method differs since our $q(\mathbf{f}|\mathbf{u})$ directly tries to construct a better approximation to the exact posterior $p(\mathbf{f}|\mathbf{u}, \mathbf{y})$, using the extra $\mathbf{V}$ variational parameters, without changing the kernel hyperparameters; see Section 3. More importantly, our method has $\mathcal{O}(NM^2)$ cost, while the ELBO in Zhu et al. (2023) (see Section 3.1 and Appendix A.1 in their paper) has cubic cost $\mathcal{O}(N^3)$ since it depends on the inverse of $\mathbf{K_{ff}} - \mathbf{Q_{ff}}$ (denoted as $\tilde{\mathbf{K}}_{nn}$ in their paper).

Artemev et al. (2021) derived an upper bound on the log determinant $\log|\mathbf{K_{ff}} + \sigma^2\mathbf{I}|$ in the exact GP log marginal likelihood and obtained the following tighter upper bound to the initial trace regularization term $-\frac{1}{2\sigma^2}\text{tr}\left(\mathbf{K_{ff}} - \mathbf{Q_{ff}}\right)$:

$$-\frac{N}{2}\log\left(1 + \frac{\text{tr}(\mathbf{K_{ff}} - \mathbf{Q_{ff}})}{N\sigma^2}\right). \qquad (24)$$

Our bound is tighter since from Jensen's inequality it holds $-\frac{N}{2}\log\left(1 + \frac{\text{tr}(\mathbf{K_{ff}} - \mathbf{Q_{ff}})}{N\sigma^2}\right) \leq -\frac{1}{2}\sum_{i=1}^{N}\log\left(1 + \frac{k_{ii} - q_{ii}}{\sigma^2}\right)$. Further, the above regularization term can be interpreted as a restricted special case of our method, obtained through a $q(\mathbf{f}|\mathbf{u})$ from Equation (13) where the diagonal matrix $\mathbf{V}$ is constrained to be spherical $\mathbf{V} = v\mathbf{I}$; see Appendix B.4. Finally note, that unlike (24) (where the sum is inside the logarithm) our bound allows to apply stochastic optimization as described in Section 3.2.

Finally, Bui et al. (2017) used power expectation propagation that minimizes $\alpha$-divergence and derived an approximation to the log marginal likelihood that interpolates between the FITC ($\alpha = 1$) log marginal likelihood (Snelson & Ghahramani, 2006; Quiñonero-Candela & Rasmussen, 2005) and the standard collapsed variational bound in (8) ($\alpha \to 0$). This approximation uses the regularization term

$$-\frac{1-\alpha}{2\alpha}\sum_{i=1}^{N}\log\left(1 + \alpha\frac{k_{ii} - q_{ii}}{\sigma^2}\right). \qquad (25)$$

This is different from ours since there is no value of $\alpha$ such that the two regularization terms will become equal. For example, note that for $\alpha \to 0$, Equation (25) reduces to $-\frac{1}{2\sigma^2}\text{tr}\left(\mathbf{K_{ff}} - \mathbf{Q_{ff}}\right)$ as discussed in Bui et al. (2017).

## 5. Experiments

### 5.1. Illustration in 1-D Regression

In the first regression experiment we consider the 1-D Snelson dataset (Snelson & Ghahramani, 2006). We took a subset of 40 examples of this dataset and we fitted the exact GP with the squared exponential kernel $k(x, x') = \sigma_f^2 \exp(-\frac{(x-x')^2}{2\ell^2})$. We also fitted sparse variational GPs with either the standard collapsed bound (Titsias, 2009) from Equation (8) (SGPR) or the new collapsed bound from Equation (16) (SGPR-new). Both sparse GP methods use seven inducing points initialized at the same values as shown in Figure 1. All methods are initialized to the same hyperparameter values; see Appendix D.

Figure 1 shows the results. Note that both SGPR and SGPR-new find similar inducing point locations. But SGPR-new, as a tighter bound (see panel (e)), is able to reduce some bias when estimating the hyperparameters since it finds a noise variance $\sigma^2$ closer to the one by exact GP (see panel (f)). This results in better predictions that match better the exact GP, as shown by the comparative visualization in panel (d). From panel (d), observe that both the mean and variances of SGPR-new are closer to the exact GP than SGPR.

### 5.2. Medium Size Regression Datasets

To further investigate the findings from the previous section, we consider three medium size real-world UCI regression datasets (Pol, Bike, and Elevators) with roughly 10k training data points each, and for which we can still run the exact GP. We choose the ARD squared exponential kernel $k(\boldsymbol{x}, \boldsymbol{x}') = \sigma_f^2 \exp(-\sum_{i=1}^{d}\frac{(x_i - x_i')^2}{2\ell_i^2})$. We run all three previous methods (Exact GP, SGPR, SGPR-new) five times with different random train-test splits; see Appendix D for experimental details. We also include in the comparison a fourth method (discussed in Related Work) which is the Artemev et al. (2021)'s bound (SGPR-artemev) that does training using the collapsed bound from Equation (36) in Appendix B.4. All sparse GP methods use $M = 1024$ or $M = 2048$ inducing points initialized by k-means. Figure 2 (in the first two lines) shows the objective function and the noise variance $\sigma^2$ across $10k$ optimization steps using Adam with base learning rate 0.01 and for $M = 1024$. For SGPR-new, the third line in Figure 2 shows histograms of the estimated final values of the optimal $v_i$ variational parameters. Figure 4 in Appendix D.1 shows the corresponding plots for $M = 2048$. We observe that for Pol and Bike, SGPR-new matches closer the exact GP training than SGPR and SGPR-artemev. Specifically, SGPR-new gives higher ELBO and estimates the noise variance with reduced underfitting bias. For the Elevators dataset, $M = 1024$ inducing points were enough for sparse GP methods to closely match exact GP training. This happens because in this case $\mathbf{Q_{ff}}$

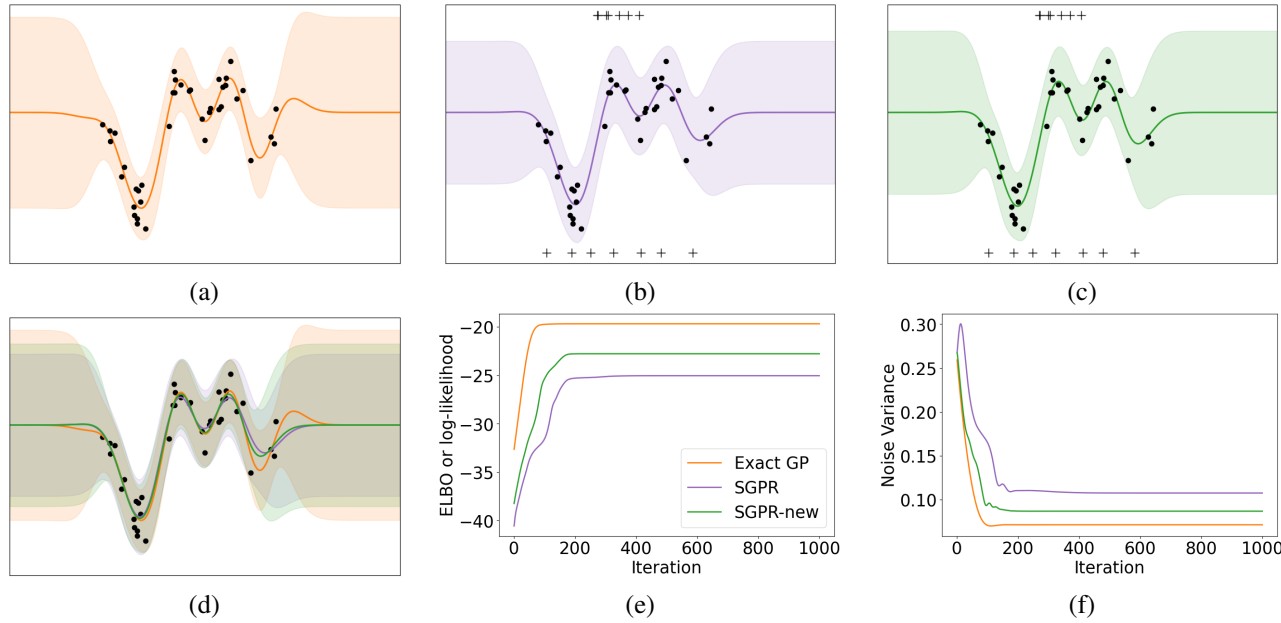

*Figure 1.* First row shows posterior predictions (means with 2-standard deviations) after fitting the exact GP (a), and the sparse GPs with either the standard collapsed SGPR bound (b) or the proposed SGPR-new collapsed bound (c). In panels (b),(c) the seven inducing points are initialized to the same random locations (shown on top with crosses) while the optimized values are shown at the bottom. Panel (d) superimposes all predictions in order to provide a more comparative visualization. Finally, panel (e) shows the ELBO (or exact log marginal likelihood for the exact GP) values across optimization steps while (f) shows the corresponding values for the noise variance $\sigma^2$.

*Table 1.* Average test log-likelihoods for the medium size regression datasets. The numbers in parentheses are standard errors.

|  | Pol | Bike | Elevators |
|---|---|---|---|
| Exact GP | 1.089(0.011) | 3.105(0.022) | −0.386(0.001) |
| $M = 1024$ |  |  |  |
| SGPR | 0.821(0.008) | 2.176(0.020) | −0.387(0.001) |
| SGPR-artemev | 0.859(0.007) | 2.199(0.024) | −0.387(0.001) |
| SGPR-new | 0.920(0.006) | 2.326(0.026) | −0.387(0.001) |
| $M = 2048$ |  |  |  |
| SGPR | 0.958(0.008) | 2.337(0.030) | −0.387(0.001) |
| SGPR-artemev | 0.976(0.008) | 2.356(0.029) | −0.387(0.001) |
| SGPR-new | 0.998(0.008) | 2.511(0.021) | −0.387(0.001) |

accurately approximates $\mathbf{K_{ff}}$, i.e., the elements $k_{ii} - q_{ii}$ get close to zero. For this latter dataset, observe that since the $k_{ii} - q_{ii}$ values are close to zero the corresponding $v_i$ values are concentrated around one as shown by the corresponding (right-most) histogram in Figure 2.

Table 1 reports test log-likelihood predictions which show that SGPR-new outperforms SGPR and SGPR-artemev.

## 5.3. Large Scale Regression Datasets

We consider 8 regression datasets, with training data sizes ranging from tens of thousands to millions. We implemented the stochastic optimization versions of the two scalable sparse GP methods: (i) the one that trains using the previous uncollapsed bound from Hensman et al. (2013) (SVGP) and (ii) our new bound from Equation (18) (SVGP-new). We denote these stochastic optimization versions by SVGP to distinguish them from the corresponding SGPR methods that use the more expensive collapsed bounds. We run the SVGP methods with $M = 1024$ and 2048 inducing points, Matern3/2 kernel with common lengthscale, minibatch size 1024, Adam with base learning rate 0.01 and 100 epochs. These experimental settings match the ones in Wang et al. (2019) and Shi et al. (2020) as further described in Appendix D.2. Table 2 reports the test log likelihood scores for all datasets. In the comparison we also included two strong baselines from Table 2 in Shi et al. (2020), i.e., SOLVE-GP and ODVGP (Salimbeni et al., 2018).

From the predictive log likelihood scores in Table 2 and also the corresponding Root Mean Squared Error (RMSE) scores reported in Table 4 in Appendix D.2, we can conclude that training with the new SVGP-new variational bound provides a clear improvement compared to training with the previous SVGP bound. Note that this improvement requires no change in the computational cost, and in fact there is

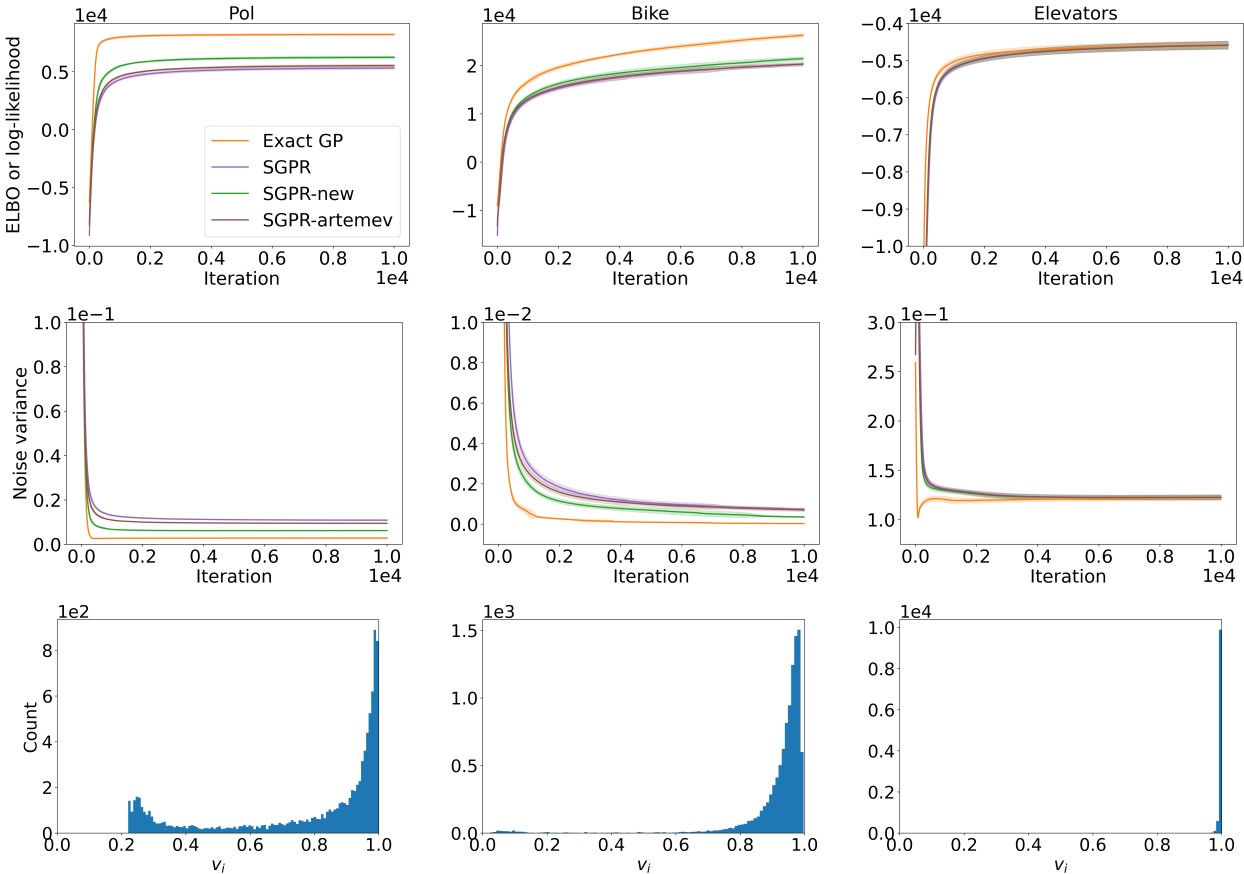

*Figure 2.* The three plots in each column correspond to the same dataset: first row shows the ELBO (or log-likelihood) for all four methods (Exact GP, SGPR, SGPR-new and SGPR-artemev) with the number of iterations, and the plot in the second row shows the corresponding values for $\sigma^2$. SGPR methods use $M = 1024$ inducing points initialized by k-means. For these two first lines we plot the mean and standard error after repeating the experiment five times with different train-test dataset splits; see Appendix D for further experimental details. For one of the runs of SGPR-new, the third line shows histograms for the estimated values of the variational parameters $v_i = \left(1 + \frac{k_{ii} - q_{ii}}{\sigma^2}\right)^{-1}$.

only a minor modification needed to be done in an existing SVGP implementation in order to run SVGP-new.

### 5.4. Poisson Regression

We consider a non-Gaussian likelihood example where the output data are counts modeled by a Poisson likelihood $p(\mathbf{y}|\mathbf{f}) = \prod_{i=1}^{N} \frac{e^{f_i}}{y_i!} e^{-e^{f_i}}$ where the log intensities values follow a GP prior. For such case the new variational approximation includes a single additional variational parameter denoted by $v$, which is optimized together with the remaining parameters; see Section 3.3. We will compare training with the new ELBO from Equation (23) (we denote this method by SVGP-new) with the standard ELBO that is obtained by restricting $v = 1$ (SVGP).

Firstly, we consider an artificial example of 50 observations with 1-D inputs placed in the grid $[-10, 10]$ where counts

are generated using Poisson intensities given by $\lambda(x) = 3.5 + 3\sin(x)$. We train the GP model with the SVGP bound and the proposed SVGP-new bound using 6 inducing points initialized to the same values for both methods; see Appendix D.3. Figure 3(left) shows the observed counts together with the predictions obtained by SVGP, SVGP-new and non-sparse variational GP (Full GP). From this figure and from the ELBO values, we observe that SVGP-new remains closer to Full GP.

Secondly, we consider a real dataset (NYBikes) about bicycles crossings going over bridges in New York City[2]. This dataset is a daily record of the number of bicycles crossing into or out of Manhattan via one of the East River bridges over a period 9 months. The data contains 210 points and

---

[2]This dataset is freely available from https://www.kaggle.com/datasets/new-york-city/nyc-east-river-bicycle-crossings.

*Table 2.* Test log-likelihoods for the large scale regression datasets with standard errors in parentheses. Best mean values are highlighted.

| | | Kin40k | Protein | KeggDirected | KEGGU | 3dRoad | Song | Buzz | HouseElectric |
|---|---|---|---|---|---|---|---|---|---|
| | $N$ | 25,600 | 29,267 | 31,248 | 40,708 | 278,319 | 329,820 | 373,280 | 1,311,539 |
| | $d$ | 8 | 9 | 20 | 27 | 3 | 90 | 77 | 9 |
| From Shi et al. (2020) | | | | | | | | | |
| ODVGP | $1024 + 1024$ | 0.137(0.003) | -0.956(0.005) | -0.199(0.067) | 0.105(0.033) | -0.664(0.003) | -1.193(0.001) | -0.078(0.001) | 1.317(0.002) |
| | $1024 + 8096$ | 0.144(0.002) | -0.946(0.005) | -0.136(0.063) | 0.109(0.033) | -0.657(0.003) | -1.193(0.001) | -0.079(0.001) | 1.319(0.004) |
| SOLVE-GP | $1024 + 1024$ | 0.187(0.002) | -0.943(0.005) | 0.973(0.003) | 0.680(0.003) | -0.659(0.002) | -1.192(0.001) | -0.071(0.001) | 1.333(0.003) |
| SVGP | 1024 | 0.108(0.002) | $-0.969(0.006)$ | 1.042(0.009) | 0.699(0.005) | $-0.704(0.003)$ | $-1.192(0.001)$ | $-0.069(0.002)$ | 1.383(0.002) |
| | 2048 | 0.237(0.002) | $-0.944(0.006)$ | **1.050**(0.009) | **0.703**(0.005) | $-$**0.650**(0.003) | $-$**1.190**(0.001) | $-0.063(0.001)$ | 1.419(0.002) |
| SVGP-new | 1024 | 0.152(0.003) | $-0.965(0.006)$ | 1.044(0.009) | 0.699(0.005) | $-0.701(0.003)$ | $-1.192(0.001)$ | $-0.065(0.002)$ | 1.387(0.003) |
| | 2048 | **0.286**(0.002) | $-$**0.938**(0.006) | 1.051(0.009) | **0.703**(0.005) | $-0.651(0.004)$ | $-$**1.190**(0.001) | $-$**0.060**(0.001) | **1.421**(0.002) |

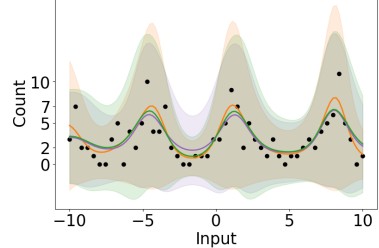
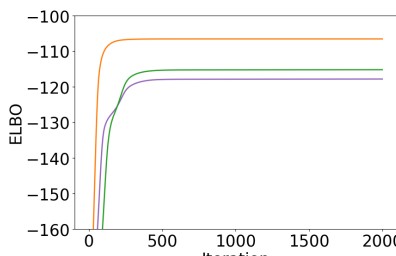
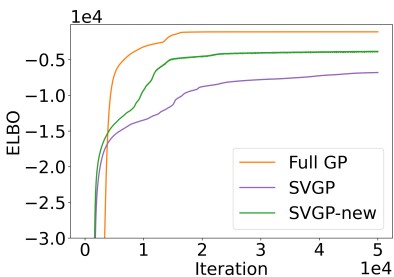

*Figure 3.* (**left**) shows the predictions (means with 2-standard deviations) over counts (black dots) in the artificial data example after fitting the Full GP, and the two SVGPs. This plot superimposes all predictions in order to provide a comparative visualization. (**middle**) shows the ELBO across optimization steps for the artificial data example. (**right**) shows the ELBO for the NYBikes dataset and $M = 16$.

we randomly choose $90\%$ for training and $10\%$ for test. We apply GP Poisson regression for the Brooklyn bridge counts where the input vector $x$ is taken to be two-dimensional consisted of maximum and minimum daily temperatures. We train the sparse GPs with either SVGP or SVGP-new and with $M = 8, 16, 32$ inducing points initialized by k-means. Since the dataset is small we also run the non-sparse Full GP. The ELBO across iterations in Figure 3 (right) and the test log likelihood scores (Table 5 in Appendix D.3) indicate that SVGP-new provides a better approximation than SVGP.

## 6. Conclusions

We have presented a method that relaxes the conditional GP assumption in the approximate distribution in sparse variational GPs. This leads to tighter collapsed and uncollapsed bounds, that maintain the computational cost with the previous bounds and can reduce training underfitting. For future work an interesting topic is to apply our method to more complex GP models, such as those with multiple outputs, with uncertain inputs and deep GPs. For the Bayesian GP-LVM, where the collapsed closed form bound has strong similarities with the previous GP regression collapsed bound in Equation (8), deriving a new collapsed bound is tractable as described in Appendix B.5. Finally, it might be useful to investigate whether theoretical convergence results on sparse GPs (Burt et al., 2020; Wild et al., 2023), can be improved given the new collapsed lower bound.

## Acknowledgements

I am grateful to the reviewers for their comments. Also, I wish to thank Jiaxin Shi and Francisco Ruiz for their invaluable advice and generous help during this project.

## Impact Statement

This paper presents work whose goal is to advance the field of Machine Learning. There are many potential societal consequences of our work, none which we feel must be specifically highlighted here.

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

# A. Further details about the SVGP method

We give a brief overview of the derivation of the standard collapsed bound in Equation (8). Some steps of the derivation will also be instructive for proving the main results of this paper in Appendix B.

Given the variational distribution $q(\mathbf{f}, \mathbf{u}) = p(\mathbf{f}|\mathbf{u})q(\mathbf{u})$ the lower bound is

$$
\begin{aligned}
\log p(\mathbf{y}) &\geq \int p(\mathbf{f}|\mathbf{u})q(\mathbf{u}) \log \frac{p(\mathbf{y}|\mathbf{f})\cancel{p(\mathbf{f}|\mathbf{u})}p(\mathbf{u})}{\cancel{p(\mathbf{f}|\mathbf{u})}q(\mathbf{u})} d\mathbf{f}d\mathbf{u} \\
&= \int p(\mathbf{f}|\mathbf{u})q(\mathbf{u}) \log \frac{p(\mathbf{y}|\mathbf{f})p(\mathbf{u})}{q(\mathbf{u})} d\mathbf{f}d\mathbf{u} \\
&= \int q(\mathbf{u}) \left\{ \int p(\mathbf{f}|\mathbf{u}) \log p(\mathbf{y}|\mathbf{f})d\mathbf{f} + \log \frac{p(\mathbf{u})}{q(\mathbf{u})} \right\} d\mathbf{u} \\
&= \int q(\mathbf{u}) \log \frac{\exp\{\int p(\mathbf{f}|\mathbf{u}) \log p(\mathbf{y}|\mathbf{f})d\mathbf{f}\}p(\mathbf{u})}{q(\mathbf{u})} d\mathbf{u}.
\end{aligned}
\tag{26}
$$

The expectation $\int p(\mathbf{f}|\mathbf{u}) \log p(\mathbf{y}|\mathbf{f})d\mathbf{f}$ can be computed as

$$
\begin{aligned}
\int p(\mathbf{f}|\mathbf{u}) \log p(\mathbf{y}|\mathbf{f})d\mathbf{f} &= \int p(\mathbf{f}|\mathbf{u}) \log p(\mathbf{y}|\mathbf{f})d\mathbf{f} \\
&= \int p(\mathbf{f}|\mathbf{u}) \left\{ -\frac{N}{2} \log(2\pi\sigma^2) - \frac{1}{2\sigma^2} \mathrm{tr} \left[ \mathbf{y}\mathbf{y}^T - 2\mathbf{y}\mathbf{f}^T + \mathbf{f}\mathbf{f}^T \right] \right\} d\mathbf{f} \\
&= -\frac{N}{2} \log(2\pi\sigma^2) - \frac{1}{2\sigma^2} \mathrm{tr} \left[ \mathbf{y}\mathbf{y}^T - 2\mathbf{y}(\mathbf{K_{fu}K_{uu}^{-1}u})^\top + (\mathbf{K_{fu}K_{uu}^{-1}u})(\mathbf{K_{fu}K_{uu}^{-1}u})^\top + \textcolor{blue}{\mathbf{K_{ff}} - \mathbf{Q_{ff}}} \right] \\
&= \log \left[ \mathcal{N}(\mathbf{y}|\mathbf{K_{fu}K_{uu}^{-1}u}, \sigma^2\mathbf{I}) \right] - \frac{1}{2\sigma^2} \mathrm{tr}(\mathbf{K_{ff}} - \mathbf{Q_{ff}}).
\end{aligned}
\tag{27}
$$

where we highlighted with blue a term in the third line to contrast it with a similar term when proving Lemma 3.2 in Appendix B.4. The ELBO in Equation (26) is written as

$$
\log p(\mathbf{y}) \geq \int q(\mathbf{u}) \log \frac{\mathcal{N}(\mathbf{y}|\mathbf{K_{fu}K_{uu}^{-1}u}, \sigma^2\mathbf{I})p(\mathbf{u})}{q(\mathbf{u})} d\mathbf{f} - \frac{1}{2\sigma^2} \mathrm{tr}(\mathbf{K_{ff}} - \mathbf{Q_{ff}}).
\tag{28}
$$

By maximizing this bound wrt the distribution $q(\mathbf{u})$ we obtain the optimal $q^*$:

$$
q^*(\mathbf{u}) = \frac{\mathcal{N}(\mathbf{y}|\mathbf{K_{fu}K_{uu}^{-1}u}, \sigma^2\mathbf{I})p(\mathbf{u})}{\int \mathcal{N}(\mathbf{y}|\mathbf{K_{fu}K_{uu}^{-1}u}, \sigma^2\mathbf{I})p(\mathbf{u})d\mathbf{u}} = \frac{\mathcal{N}(\mathbf{y}|\mathbf{K_{fu}K_{uu}^{-1}u}, \sigma^2\mathbf{I})p(\mathbf{u})}{\mathcal{N}(\mathbf{y}|\mathbf{0}, \mathbf{Q_{ff}} + \sigma^2\mathbf{I})}
\tag{29}
$$

$$
= \mathcal{N}(\mathbf{u}|\sigma^{-2}\mathbf{K_{uu}}(\mathbf{K_{uu}} + \sigma^{-2}\mathbf{K_{uf}K_{fu}})^{-1}\mathbf{K_{uf}y}, \mathbf{K_{uu}}(\mathbf{K_{uu}} + \sigma^{-2}\mathbf{K_{uf}K_{fu}})^{-1}\mathbf{K_{uu}}),
\tag{30}
$$

where the expression in the second line (obtained after some standard completion of a square procedure) shows that $q^*(\mathbf{u})$ can be computed in $\mathcal{O}(NM^2)$ time. In fact, this optimal $q^*(\mathbf{u})$ is the same as the one obtained by the DTC (also known as projected process) approximation (Seeger et al., 2003; Quiñonero-Candela & Rasmussen, 2005). By substituting the expression in (29) into the bound in (28) we obtain the well-known formula of the collapsed bound:

$$
\log p(\mathbf{y}) \geq \log \mathcal{N}(\mathbf{y}|\mathbf{0}, \mathbf{Q_{ff}} + \sigma^2\mathbf{I}) - \frac{1}{2\sigma^2} \mathrm{tr}(\mathbf{K_{ff}} - \mathbf{Q_{ff}}).
\tag{31}
$$

Given the Gaussian form of $q(\mathbf{u}) = \mathcal{N}(\mathbf{u}|\boldsymbol{\mu}, \mathbf{A})$ the posterior GP is given by $q(\mathbf{f}_*) = \int p(\mathbf{f}_*|\mathbf{u})q(\mathbf{u})d\mathbf{u}$:

$$
q(\mathbf{f}_*) = \mathcal{N}(\mathbf{f}_*|\mathbf{K_{f_*f}K_{uu}^{-1}}\boldsymbol{\mu}, \mathbf{K_{f_*f_*}} - \mathbf{K_{f_*u}K_{uu}^{-1}K_{uf_*}} + \mathbf{K_{f_*u}K_{uu}^{-1}AK_{uu}^{-1}K_{uf_*}})
\tag{32}
$$

which further simplifies if we substitute the optimal mean and covariance of $q^*(\mathbf{u})$:

$$
q(\mathbf{f}_*) = \mathcal{N}(\mathbf{f}_*|\mathbf{K_{f_*f}\Lambda^{-1}K_{uf}} \frac{\mathbf{y}}{\sigma^2}, \mathbf{K_{f_*f_*}} - \mathbf{K_{f_*u}K_{uu}^{-1}K_{uf_*}} + \mathbf{K_{f_*u}\Lambda^{-1}K_{uf_*}})
\tag{33}
$$

where $\mathbf{\Lambda} = \mathbf{K_{uu}} + \sigma^{-2}\mathbf{K_{uf}K_{fu}}$.

# B. Further details about the proposed bounds

Here, we provide several proofs regarding the proposed bounds.

## B.1. Proof of Lemma 3.1

$q(\mathbf{f}|\mathbf{u})$ and $p(\mathbf{f}|\mathbf{u})$ are Gaussian distributions having the same mean but different covariance matrices. Thus, the KL divergence reduces to

$$\mathrm{KL}[q(\mathbf{f}|\mathbf{u})||p(\mathbf{f}|\mathbf{u})] = \frac{1}{2} \left\{ \log \frac{|\mathbf{K_{ff}} - \mathbf{Q_{ff}}|}{|(\mathbf{K_{ff}} - \mathbf{Q_{ff}})^{\frac{1}{2}} \mathbf{V} (\mathbf{K_{ff}} - \mathbf{Q_{ff}})^{\frac{1}{2}}|} - N + \mathrm{tr}\{(\mathbf{K_{ff}} - \mathbf{Q_{ff}})^{-1} (\mathbf{K_{ff}} - \mathbf{Q_{ff}})^{\frac{1}{2}} \mathbf{V} (\mathbf{K_{ff}} - \mathbf{Q_{ff}})^{\frac{1}{2}}\} \right\}$$

$$= \frac{1}{2} \left\{ -\log|\mathbf{V}| - N + \mathrm{tr}\{\mathbf{V}\} \right\}, \tag{34}$$

where all the terms involving $\mathbf{K_{ff}} - \mathbf{Q_{ff}}$ cancel out by using standard properties of the matrix determinant and trace. Now since $\mathbf{V}$ is a diagonal matrix (with diagonal elements $v_i > 0$) we conclude that $\mathrm{KL}[q(\mathbf{f}|\mathbf{u})||p(\mathbf{f}|\mathbf{u})] = \frac{1}{2} \sum_{i=1}^{N} (v_i - \log v_i - 1)$.

## B.2. Proof of Lemma 3.2

The derivation of $\int q(\mathbf{f}|\mathbf{u}) \log p(\mathbf{y}|\mathbf{f}) d\mathbf{f}$ is similar to the derivation in Equation (27) with a small difference highlighted in blue:

$$\int p(\mathbf{f}|\mathbf{u}) \log p(\mathbf{y}|\mathbf{f}) d\mathbf{f}$$

$$= -\frac{N}{2} \log(2\pi\sigma^2) - \frac{1}{2\sigma^2} \mathrm{tr} \left[ \mathbf{y}\mathbf{y}^T - 2\mathbf{y}(\mathbf{K_{fu}K_{uu}^{-1}u})^\top + (\mathbf{K_{fu}K_{uu}^{-1}u})(\mathbf{K_{fu}K_{uu}^{-1}u})^\top + (\mathbf{K_{ff}} - \mathbf{Q_{ff}})^{\frac{1}{2}} \mathbf{V} (\mathbf{K_{ff}} - \mathbf{Q_{ff}})^{\frac{1}{2}} \right]$$

$$= \log \left[ \mathcal{N}(\mathbf{y}|\mathbf{K_{fu}K_{uu}^{-1}u}, \sigma^2\mathbf{I}) \right] - \frac{1}{2\sigma^2} \mathrm{tr}(\mathbf{V}(\mathbf{K_{ff}} - \mathbf{Q_{ff}})), \tag{35}$$

where we used that $\mathrm{tr}((\mathbf{K_{ff}} - \mathbf{Q_{ff}})^{\frac{1}{2}} \mathbf{V} (\mathbf{K_{ff}} - \mathbf{Q_{ff}})^{\frac{1}{2}}) = \mathrm{tr}(\mathbf{V}(\mathbf{K_{ff}} - \mathbf{Q_{ff}}))$. Now since $\mathbf{V}$ is a diagonal matrix we have $\mathrm{tr}(\mathbf{V}(\mathbf{K_{ff}} - \mathbf{Q_{ff}})) = \sum_{i=1}^{N} v_i(k_{ii} - q_{ii})$ which completes the proof.

## B.3. Proof of Proposition 3.3

The ELBO is written as

$$\log p(\mathbf{y}) \geq \int q(\mathbf{f}|\mathbf{u})q(\mathbf{u}) \log \frac{p(\mathbf{y}|\mathbf{f})p(\mathbf{f}|\mathbf{u})p(\mathbf{u})}{q(\mathbf{f}|\mathbf{u})q(\mathbf{u})} = \int q(\mathbf{u}) \left\{ \log \frac{\exp\{\mathbb{E}_{q(\mathbf{f}|\mathbf{u})}[\log p(\mathbf{y}|\mathbf{f})]\}p(\mathbf{u})}{q(\mathbf{u})} - \mathrm{KL}[q(\mathbf{f}|\mathbf{u})||p(\mathbf{f}|\mathbf{u})] \right\} d\mathbf{u}$$

and by using the results from the two lemmas this becomes

$$\log p(\mathbf{y}) \geq \int q(\mathbf{u}) \log \frac{\mathcal{N}(\mathbf{y}|\mathbf{K_{fu}K_{uu}^{-1}u}, \sigma^2\mathbf{I})p(\mathbf{u})}{q(\mathbf{u})} d\mathbf{u} - \frac{1}{2} \sum_{i=1}^{N} \left\{ v_i \left( 1 + \frac{k_{ii} - q_{ii}}{\sigma^2} \right) - \log v_i - 1 \right\}.$$

Clearly, maximizing over $q(\mathbf{u})$ gives the same optimal distribution as in Equation (29), and the first term in the bound is the DTC log likelihood. The second term that depends on the $v_i$s is a concave function over these parameters. Thus, by differentiating and setting to zero we obtain the optimal values $v_i^* = \left( 1 + \frac{k_{ii} - q_{ii}}{\sigma^2} \right)^{-1}$. If we plug these values back into the bound we obtain the new tighter collapsed bound in Proposition 3.3.

## B.4. Reinterpretation of Artemev et al. (2021)'s bound

We consider the following form of $q(\mathbf{f}|\mathbf{u})$:

$$q(\mathbf{f}|\mathbf{u}) = \mathcal{N}(\mathbf{f}|\mathbf{K_{fu}K_{uu}^{-1}u}, v(\mathbf{K_{ff}} - \mathbf{Q_{ff}})).$$

Then, $\mathrm{KL}[q(\mathbf{f}|\mathbf{u})||p(\mathbf{f}|\mathbf{u})] = \frac{N}{2}(v - \log v - 1)$ and $\mathbb{E}_{q(\mathbf{f}|\mathbf{u})}[\log p(\mathbf{y}|\mathbf{f})]$
$= \log \mathcal{N}(\mathbf{y}|\mathbf{K_{fu}K_{uu}^{-1}u}, \sigma^2\mathbf{I}) - \frac{v}{2\sigma^2}\mathrm{tr}(\mathbf{K_{ff}} - \mathbf{Q_{ff}})$ and the bound is written as

$$\log p(\mathbf{y}) \geq \int q(\mathbf{u}) \log \frac{\mathcal{N}(\mathbf{y}|\mathbf{K_{fu}K_{uu}^{-1}u}, \sigma^2\mathbf{I})p(\mathbf{u})}{q(\mathbf{u})} d\mathbf{u} - \frac{1}{2} \left\{ \frac{v}{\sigma^2}\mathrm{tr}(\mathbf{K_{ff}} - \mathbf{Q_{ff}}) + N(v - \log v - 1) \right\}.$$

By maximizing wrt $v$ we obtain $v^* = \left(1 + \frac{\text{tr}(\mathbf{K_{ff}} - \mathbf{Q_{ff}})}{N\sigma^2}\right)^{-1}$, and by substituting this back into the bound we obtain Artemev et al. (2021)'s tighter bound on the initial trace regularization term. Overall this collapsed bound has the form

$$\log p(\mathbf{y}) \geq \log \mathcal{N}(\mathbf{y}|\mathbf{0}, \mathbf{Q_{ff}} + \sigma^2 \mathbf{I}) - \frac{N}{2}\log\left(1 + \frac{\text{tr}(\mathbf{K_{ff}} - \mathbf{Q_{ff}})}{N\sigma^2}\right). \tag{36}$$

This collapsed bound is what the method SGPR-artemev is using in Section 5.2. Note that Artemev et al. (2021) propose also additional but more expensive bounds for the first DTC log likelihood term that require running conjugate gradients. We do not consider those in our comparisons (such bounds could be used in all SGPR bounds since all share the same DTC log likelihood term) as they have higher cost.

### B.5. Bound for the Bayesian GP-LVM

Due to the strong similarity of the standard collapsed SVGP bound in Equation (8) with the collapsed bound in the Bayesian GP-LVM (Titsias & Lawrence, 2010), applying the new approximation to Bayesian GP-LVM seems to be simple and we discuss it next.

Given observed data $Y \in \mathbb{R}^{N \times D}$ and latent variables $X \in \mathbb{R}^{N \times Q}$ we have the latent variable model

$$p(Y|X)p(X) = \left(\prod_{d=1}^{D} p(\mathbf{y}_d|X)\right)p(X),$$

where $p(X)$ is a Gaussian prior over the latent variables and $p(\mathbf{y}_d|X) = \mathcal{N}(\mathbf{y}_d|\mathbf{K_{ff}}(X) + \sigma^2 \mathbf{I})$. Given a Gaussian variational distribution $q(X)$ over the latent variables, the initial form of the bound is

$$F = \int q(X) \log p(Y|X)dX - \text{KL}[q(X)||p(X)]$$

$$= \sum_{d=1}^{D} \int q(X) \log p(\mathbf{y}_d|X)dX - \text{KL}[q(X)||p(X)]$$

$$= \sum_{d=1}^{D} F_d - \text{KL}[q(X)||p(X)], \tag{37}$$

where $F_d = \int q(X) \log p(\mathbf{y}_d|X)dX$. The KL part will be a tractable KL between two Gaussians, and thus the difficulty is to approximate $F_d$. Given that $\log p(\mathbf{y}_d|X)$ has the same form with the log marginal likelihood in GP regression, we can lower bound it using inducing variables and exactly the same form of $q(\mathbf{f}|\mathbf{u})$ as we did in the main paper. This gives

$$\log p(\mathbf{y}_d|X) \geq \int q(\mathbf{u}) \log \frac{\mathcal{N}(\mathbf{y}_d|\mathbf{K_{fu}}(X)\mathbf{K_{uu}^{-1}}\mathbf{u}, \sigma^2 \mathbf{I})p(\mathbf{u})}{q(\mathbf{u})}d\mathbf{u} - \frac{1}{2}\sum_{i=1}^{N}\left\{v_i\left(1 + \frac{k(\boldsymbol{x}_i, \boldsymbol{x}_i) - q(\boldsymbol{x}_i, \boldsymbol{x}_i)}{\sigma^2}\right) - \log v_i - 1\right\},$$

where $\boldsymbol{x}_i$ is the latent variable for the $i$-th data point and $q(\boldsymbol{x}_i, \boldsymbol{x}_i) = \mathbf{k}(x_i)^\top \mathbf{K_{uu}^{-1}} \mathbf{k}(\boldsymbol{x}_i) = \text{tr}\{\mathbf{K_{uu}^{-1}}\mathbf{k}(\boldsymbol{x}_i)\mathbf{k}(\boldsymbol{x}_i)^\top\}$. Note that we write the cross kernel matrix as $\mathbf{K_{fu}}(X)$ to emphasize its dependence on the latent variables $X$, while $\mathbf{K_{uu}}$ does not depend on $X$. Note also that we assume that each $v_i$ parameter does not depend on $\boldsymbol{x}_i$ and this is crucial to obtain a closed form collapsed bound. Now we follow the derivation in the initial Bayesian GP-LVM where we use the above bound to replace $\log p(\mathbf{y}_d|X)$ in $\int q(X) \log p(\mathbf{y}_d|X)dX$ and do first the expectation over $X$, and then solve for the optimal $q(\mathbf{u})$. This eliminates $q(\mathbf{u})$ and it gives the bound

$$\log \int e^{\langle \mathcal{N}(\mathbf{y}_d|\mathbf{K_{fu}}(X)\mathbf{K_{uu}^{-1}}\mathbf{u}, \sigma^2 \mathbf{I})\rangle_{q(X)}}p(\mathbf{u})d\mathbf{u} - \frac{1}{2}\sum_{i=1}^{N}\left\{v_i\left(1 + \frac{\langle k(\boldsymbol{x}_i, \boldsymbol{x}_i)\rangle_{q(\boldsymbol{x}_i)} - \langle q(\boldsymbol{x}_i, \boldsymbol{x}_i)\rangle_{q(\boldsymbol{x}_i)}}{\sigma^2}\right) - \log v_i - 1\right\}.$$

where we have used physics notation for expectation, i.e., $\langle \cdot \rangle$. For $v_i = 1$ this is the previous collapsed bound used by Bayesian GP-LVM. By maximizing over each $v_i$ we obtain the new collapsed bound

$$F_d \geq \log \int e^{\langle \mathcal{N}(\mathbf{y}_d|\mathbf{K_{fu}}(X)\mathbf{K_{uu}^{-1}}\mathbf{u}, \sigma^2 \mathbf{I})\rangle_{q(X)}}p(\mathbf{u})d\mathbf{u} - \frac{1}{2}\sum_{i=1}^{N}\log\left(1 + \frac{\langle k(\boldsymbol{x}_i, \boldsymbol{x}_i)\rangle_{q(\boldsymbol{x}_i)} - \langle q(\boldsymbol{x}_i, \boldsymbol{x}_i)\rangle_{q(\boldsymbol{x}_i)}}{\sigma^2}\right),$$

which can be substituted in the overall Bayesian GP-LVM bound above. Again the implementation of the new bound requires a minor modification to existing Bayesian GP-LVM code.

## C. Learned hyperparameters in 1-D Snelson dataset

Table 3 provides the learned hyperparameters for the 1-D Snelson dataset.

Table 3. Hyperparameter values in 1-D Snelson example.

|          | $\sigma^2$ | $\sigma_f^2$ | $\ell^2$ |
| -------- | ------ | ------ | ----- |
| Exact GP | 0.0715 | 0.712  | 0.597 |
| SVGP-new | 0.087  | 0.485  | 0.615 |
| SVGP     | 0.108  | 0.331  | 0.617 |

## D. Further experimental details and results

For all regression experiments (apart from the toy Snelson 1-D dataset) we repeat the runs for five times using different random training and test splits. By following Wang et al. (2019) and Shi et al. (2020) we consider 80% / 20% training / test splits. A 20% subset of the training set is used for validation.

The training inputs and regression outputs are normalized to have zero mean. For the hyperparameters $\sigma^2, \sigma_f^2, \ell^2$ (or $\ell_i^2$ for ARD kernels) we use the softplus activation to parametrize the square roots of these parameters, i.e., to parametrize $\sigma, \sigma_f, \ell_i$. For all experiments we use the initializations $\sigma = 0.51, \sigma_f = 0.69, \ell_i = 1.0$. The inducing inputs $\mathbf{Z}$ are initialized by running at maximum 30 iterations of k-means clustering with the centers initialized at a random training data subset.

### D.1. Medium size regression datasets

For all three datasets in this section we can run Exact GP given the medium training size. For the Pol dataset the training size is $N = 9600$ and input dimensionality $d = 26$. For Elevators is $N = 10623$ and $d = 18$. For the Bike dataset the initial train size (see e.g., Table 7 in Shi et al. (2020)) is $N = 11122$ (with $d = 17$) but since Exact GP training gave out-of-memory error when running in a V100 GPU, we had to slightly reduce the training size to $N = 10600$.

An mentioned in the main paper the standard squared exponential ARD kernel was used in all experiments in this section. For training, we perform 10000 optimization iterations using the Adam optimizer with base learning 0.01.

Figure 4 shows the objective function values and noise variance parameter $\sigma^2$ across iterations when the SGPR methods use $M = 2048$ inducing points. Figure 2 in the main paper shows the result for $M = 1024$.

### D.2. Large scale regression datasets

The experimental settings are chosen to match the ones from Wang et al. (2019) and Shi et al. (2020), where we used GPs with a Matérn32 kernel (with common lengthscale). Following these settings, for all datasets we train for 100 epochs using Adam with learning rate 0.01 and minibatch size 1024.

Table 4 reports RMSE scores, while test log likelihood scores are given in Table 2 of the main paper.

### D.3. Poisson regression

Figure 5 shows the ELBOs across iterations for the NYBikes dataset for $M = 8$ and $M = 32$ inducing points, while the plot for $M = 16$ is shown in the main paper. Table 5 presents test log-likelihood scores for the NYBikes dataset. Average values and standard errors are computed by repeating the experiment five times where at each repeat we randomly split the initial data into 90% for training and 10% for test.

Regarding the scalar value of $v$ for the toy Poisson regression the learned value was around $v = 0.675$, while or the real Poisson example in NYBikes, the value gets very small below 0.01.

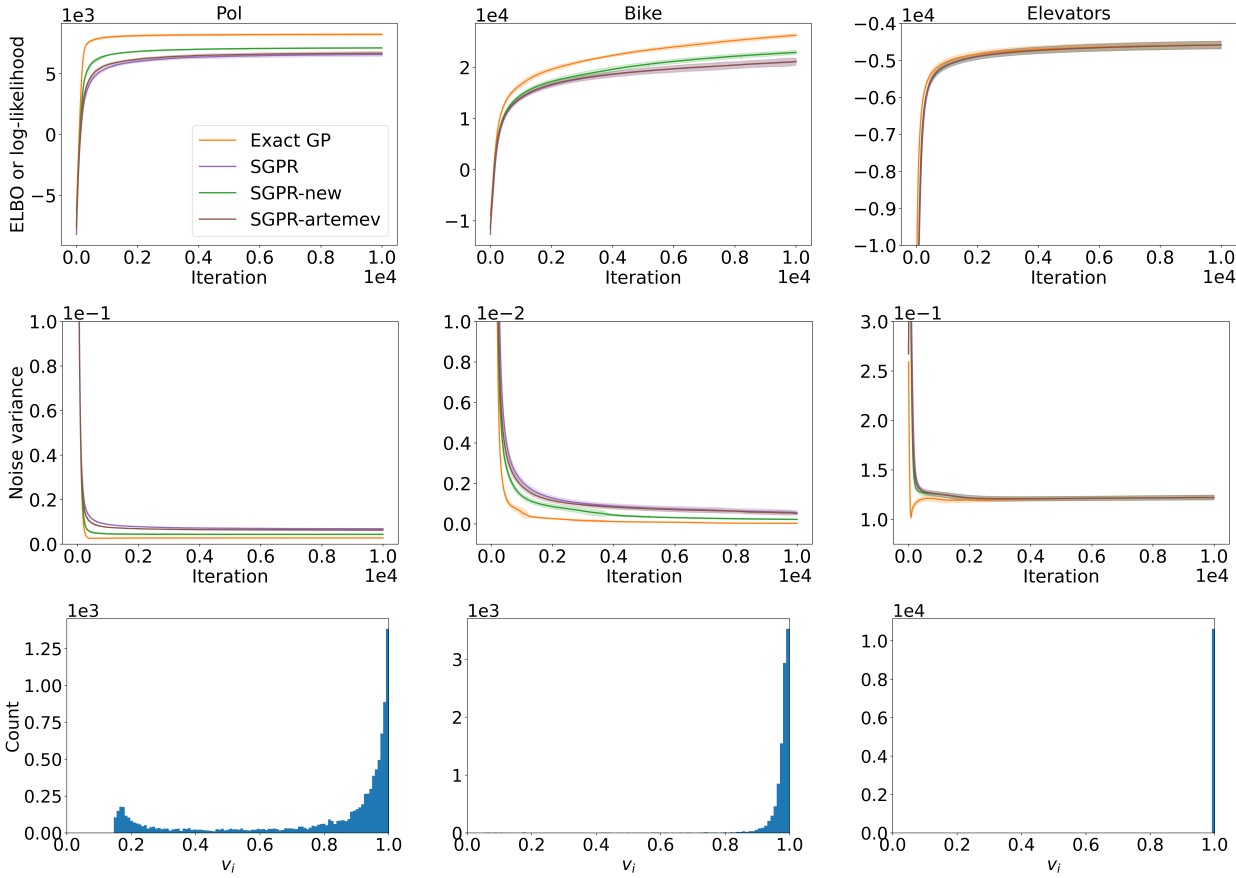

*Figure 4.* The two plots in each column correspond to the same dataset: first row shows the ELBO (or log-likelihood) for all four methods (Exact GP, SGPR, SGPR-new and SGPR-artemev) with the number of iterations and the plot in the second row shows the corresponding values for $\sigma^2$. SGPR methods use $M = 2048$ inducing points initialized by k-means. For these two first lines we plot the mean and standard error after repeating the experiment five times with different train-test dataset splits. For one of the runs of SGPR-new, the third line shows histograms for the estimated values of the variational parameters $v_i$.

*Table 4.* Test RMSE values of large scale regression datasets with standard errors in parentheses. Best mean values are highlighted.

|  |  | Kin40k | Protein | KeggDirected | KEGGU | 3dRoad | Song | Buzz | HouseElectric |
|---|---|---|---|---|---|---|---|---|---|
|  | $N$ | 25,600 | 29,267 | 31,248 | 40,708 | 278,319 | 329,820 | 373,280 | 1,311,539 |
|  | $d$ | 8 | 9 | 20 | 27 | 3 | 90 | 77 | 9 |
| From Shi et al. (2020) |  |  |  |  |  |  |  |  |  |
| ODVGP | $1024 + 1024$ | 0.183(0.001) | 0.625(0.004) | 0.176(0.012) | 0.156(0.004) | 0.467(0.001) | 0.797(0.001) | 0.263(0.001) | 0.062(0.000) |
|  | $1024 + 8096$ | 0.180(0.001) | 0.618(0.004) | 0.157(0.009) | 0.157(0.004) | 0.462(0.002) | 0.797(0.001) | 0.263(0.001) | 0.062(0.000) |
| SOLVE-GP | $1024 + 1024$ | 0.172(0.001) | 0.618(0.004) | 0.095(0.002) | 0.123(0.001) | 0.464(0.001) | 0.796(0.001) | 0.261(0.001) | 0.061(0.000) |
| SVGP | 1024 | 0.195(0.001) | 0.635(0.004) | 0.086(0.001) | 0.122(0.001) | 0.486(0.002) | 0.797(0.001) | 0.261(0.001) | 0.059(0.000) |
|  | 2048 | 0.171(0.000) | 0.619(0.004) | 0.086(0.001) | **0.121**(0.001) | **0.460**(0.002) | **0.795**(0.001) | 0.260(0.001) | **0.057**(0.000) |
| SVGP-new | 1024 | 0.182(0.001) | 0.631(0.004) | 0.086(0.001) | 0.122(0.001) | 0.484(0.001) | 0.796(0.001) | 0.259(0.001) | 0.058(0.000) |
|  | 2048 | **0.158**(0.000) | **0.615**(0.004) | 0.086(0.001) | **0.121**(0.001) | 0.461(0.002) | **0.795**(0.001) | **0.258**(0.000) | **0.057**(0.000) |

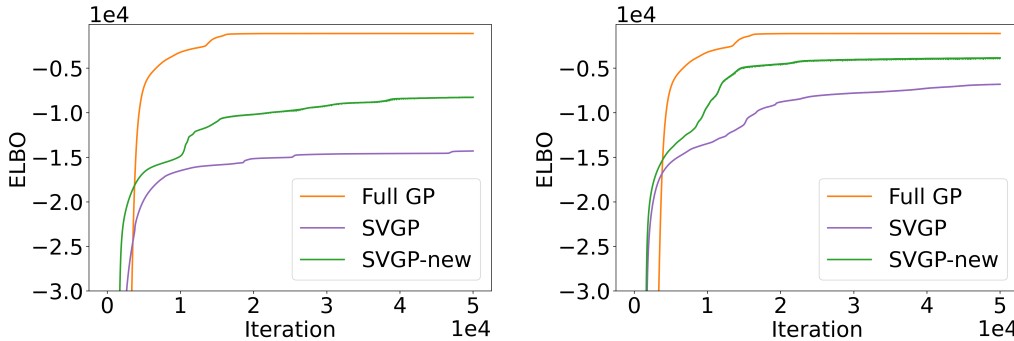

*Figure 5.* The left panel shows the lower bounds across iterations when the sparse GP methods run with $M = 8$ inducing points, while the right panel shows the corresponding plot with $M = 32$ inducing points.

*Table 5.* Test log likelihoods on the NYBikes Poisson regression dataset with standard errors in parentheses. For the sparse methods we consider varying numbers of inducing points, i.e., $M = 8, 16, 32$.

| | | |
|---|---|---|
| Full GP | | $-5.061(0.010)$ |
| SVGP | 8 | $-36.397(6.017)$ |
| SVGP | 16 | $-16.557(4.307)$ |
| SVGP | 32 | $-8.556(0.728)$ |
| SVGP-new | 8 | $-9.713(0.345)$ |
| SVGP-new | 16 | $-9.301(0.296)$ |
| SVGP-new | 32 | $-8.203(0.190)$ |

