# OpenReview forum: "New Bounds for Sparse Variational Gaussian Processes"
_ICML.cc/2025/Conference — ICML 2025 spotlightposter_

### Official Review · Reviewer_1vBe · 2025-03-09

**Overall Recommendation:** 4

**Summary:**

The authors introduce a tighter ELBO bound for inducing-point-based Gaussian process regression à la SVGP (Titsias 2009) and its mini-batchable extension (Hensman et al. 2013). The main idea is to use a more flexible ansatz for $q({\bf f} | {\bf u})$ than the conditional prior $p({\bf f} | {\bf u})$, in particular by introducing $N$ variational parameters into the covariance of $q({\bf f} | {\bf u})$, where $N$ is the total number of training data points. This leads to a small change in the form of the ELBO while retaining the same computational cost as SVGP and retaining mini-batchability in the case of the tightened Hensman-et-al-like objective (i.e. in the case where $\bf u$ is not integrated out optimally). The authors provide a careful discussion of how the new bound differs from previous alternative bounds. The authors also consider how this construction can be applied to the setting with a non-gaussian likelihood. Experiments suggest that the new method can lead to higher ELBOs and better predictive distributions in practice.

**Claims And Evidence:**

The main claims made by the authors are that their method can:

1. reduce bias when learning the hyperparameters of the kernel
2. can lead to better predictive performance
3. result in tighter ELBOs

I would say that the evidence for 2 and 3 is convincing, while the evidence for 1 is somewhat weak. If this is to be a central claim of the submission, it should be supported with more empirical evidence, in particular in the simulated data setting where "bias" has a particularly straightforward meaning.

**Essential References Not Discussed:**

N/A

**Experimental Designs Or Analyses:**

The setup of the experiments seems to follow best practices and otherwise looks sound. (Though personally I would never use a fixed Adam learning rate of $0.01$ without decaying it at least to some extent, as this is generically unlikely to result in particularly well-optimized objectives).

**Methods And Evaluation Criteria:**

While the empirical evaluation is not extraordinarily extensive, it is enough to convince me of the basic soundness and advantages of the proposed approach. Nevertheless, I would love to see the experiments and/or reported results extended in a few directions:
- The authors note the tendency of SVGP to overestimate the observation noise and provide hints that the new objective lessens this bias. It would be great to see a more extensive study of this particular question, especially on simulated data, and reporting more detailed results for other kernel hyperparameters (e.g. the learned kernel scale).
- I would love to get more details on the nature of the learned $v$s. Can the authors report a histogram and/or summary statistics for a few sample cases? Similarly in the non-gaussian case where $v$ is a scalar, what values of $v$ are you finding in practice? It's perhaps somewhat surprising to me that a single additional degree of freedom in the variational distribution can have the moderately large effect we can read off from Figure 3 (though in truth without predictive NLLs and the like it's a bit hard to measure the magnitude of the improvement/effect).

**Other Comments Or Suggestions:**

- typo in abstract: "hyperpaparameters"
- typo in sec 3.2: whiten => whitened
- typo: "intiliazed" in Fig 1

**Other Strengths And Weaknesses:**

I would like to again stress the great clarity of the manuscript and thank the authors for taking the time to write a very clear and well-argued manuscript.

**Questions For Authors:**

Have you considered adding a new $N$-dimensional "mean shift" variational parameter to your ansatz for $q({\bf f } | {\bf u})$? In particular this would mean using a mean of $\bf K_{fu} K^{-1}_{uu} u + \Delta$ for ${\bf \Delta} \in \mathbb{R}^N$. This would seem to be a natural companion to your adjustment of the covariance matrix. And the exact form for $p({\bf f}|{\bf u}, {\bf y})$ in Sec. 3 suggests that there should be some slackness in the ELBO resulting from using the prior conditional mean without any modification. Granted this new parameter might conceivably be difficult to optimize (say because its entangled with the variational mean for ${\bf u}$), or such a modification may have little effect on ELBO tightness or the learned hyperparameters or the learned predictive distribution, but from a methodological point of view it seems a very natural question to ask, so its omission is unfortunate. Why not satisfy the reader's curiosity?

**Relation To Broader Scientific Literature:**

The discussion of related work and how this new bound fits into the existing methodological landscape is extensive, very clear, and generally excellent.

**Theoretical Claims:**

I have not checked the derivations in detail, but they are all intuitive/non-surprising and as such I have no reason to doubt their correctness.

---

> ### Author Rebuttal · Authors · 2025-03-31
>
> Thank you for the insightful comments.
>
> > I would say that the evidence for 2 and 3 is convincing, while the evidence for 1 is somewhat weak. If this is to be a central claim of the submission, it should be supported with more empirical evidence, in particular in the simulated data setting where "bias" has a particularly straightforward meaning.
>
> > The authors note the tendency of SVGP to overestimate the observation noise and provide hints that the new objective lessens this bias. It would be great to see a more extensive study of this particular question, especially on simulated data, and reporting more detailed results for other kernel hyperparameters (e.g. the learned kernel scale).
>
> We provide  here the learned hyperparameters for the 1-D Snelson dataset:
>
>             | Noise Variance  | Scale/amplitude  | Lengthscale|
>
> Exact GP    | 0.0715    | 0.712    | 0.597 |
>
> SVGP-new | 0.087     | 0.485      | 0.615|
>
> SVGP         | 0.108       | 0.331 |  0.617 |
>
> which show that SVGP-new has less bias than SVGP in this example. We plan to add further simulated regression examples
> in the appendix by following Reviewer's suggestion.
>
>
> > I would love to get more details on the nature of the learned vs. Can the authors report a histogram and/or summary statistics
>
> Yes, we can add histograms for the learned $v_i$s for some of the GP regression datasets (e.g. Snelson, Pol, Bike, Elevators) in the appendix. We give here some summary statistics (min, median and max values) of the learned $v_i$s  for the 1-D Snelson example (from  Figure 1):
>
> |min |  median | max |
>
> |0.172 |  0.952 |  0.9998 |
>
> This indicates that there are few $v_i$s having quite small values but most of them are close to one in this example. We will update the Appendix to add histograms that visualize the final values of $v_i$s.
>
> Regarding the scalar values of $v$ for the non-Gaussian Poisson regression runs, we report here the
> learned values.  For the toy Poisson regression the learned value was around $v=0.675$. Interestingly, for the real Poisson example in NYBikes, the value gets very small below $0.01$.  Perhaps this suggests that in this example the expected Poisson log-likelihood reconstruction term has a very strong influence in ELBO optimization,  so that the optimization prefers to set $v$ very small (which allows to reduce the posterior variance in $q(f_i)$ since the term $v( k_{ii} - q_{ii})$ becomes small). If the number of inducing points becomes sufficiently large and each $k_{ii} - q_{ii}$ becomes small, then of course the learned $v$ will become close to 1.
>
> | Have you considered adding a new $N$-dimensional "mean shift"
>
> This is a great suggestion. Yes, we tried to do this but we were unable to find to way to add a drift vector $\Delta$ that gives a computational efficient ELBO of $O(N M^2)$ cost.  Here, is a short derivation of this. In the KL divergence $KL[q(f|u) || p(f|u)]$  and since the $q(f|u)$ and $p(f|u)$ do not have the same mean anymore the term $\frac{1}{2} \Delta^\top (K_{f f} - Q_{f f })^{-1} \Delta$ appears which has $O(N^3)$ cost. If we try to add some preconditioning to $\Delta$, e.g., $(K_{ff } - Q_{ff})^{1/2} \Delta$ then the $O(N^3)$ remains (although this time will appear in the expected log-likelihood term). We can add an appendix about these derivations, since they could be useful for future research.

---

### Official Review · Reviewer_Pyzb · 2025-03-10

**Overall Recommendation:** 3

**Summary:**

The paper revisits the widely-used variational approximation for sparse Gaussian processes (GPs). It proposes a refined variational formulation, introducing a more flexible conditional posterior distribution in place of the traditional assumption (where the conditional posterior matches the prior). This adjustment results in a tighter variational bound on the marginal likelihood, improving inference quality. Additionally, the method naturally supports stochastic mini-batch optimization, making it scalable to large datasets and practical for a broader range of applications.

**Claims And Evidence:**

Yes, the claims made in the submission are supported by clear and convincing evidence. The paper provides a well-structured theoretical foundation for the proposed variational approximation, including a derivation that justifies the tighter variational bound.

**Essential References Not Discussed:**

There is another paper that was released at a similar time that essentially uses the same formulation and results in very similar findings. While I understand that this work can be considered concurrent, it would be beneficial for the authors to acknowledge and discuss this related research. Including a mention of this paper would provide readers with a clearer understanding of how the contributions of this work fit into the broader context and help differentiate any unique aspects of the approach.

Tighter sparse variational Gaussian processes, Bui et al (2025), Under review TMLR.

**Experimental Designs Or Analyses:**

NA

**Methods And Evaluation Criteria:**

Yes, mainly. The proposed methods and evaluation criteria are appropriate for the problem at hand, and the experiments effectively demonstrate the benefits of the tighter variational bound. However, the datasets used in the evaluation seem somewhat simple, and the paper could have further strengthened its empirical validation by including a more complex benchmark, such as MNIST or a real-world dataset. This would provide additional evidence of the method’s scalability and effectiveness in practical applications.

**Other Comments Or Suggestions:**

- Table 2 is confusing. The top methods are baselines but have nothing to do with your method I assume. The bottom methods both have ours but isn't your method only SVGP-new?

**Other Strengths And Weaknesses:**

The main strength of the paper lies in its effort to improve the approximation of sparse variational Gaussian processes (SVGP), a crucial technique for scaling Gaussian processes. Additionally, the experimental results on benchmark datasets are compelling, providing strong empirical support for the proposed method. The paper is well-written, and the core idea is clearly explained.

The main weakness of the paper is that the results demonstrate only marginal improvements with the new bound across all evaluated tasks, without a clear example where the standard approach would fail without it. This makes it difficult to assess the practical necessity of the proposed refinement. Additionally, in the non-Gaussian likelihood setting, the paper opts for a simpler approximation rather than leveraging the more expressive variational form, seemingly due to computational constraints.

**Questions For Authors:**

- When moving to the minibatch setting, the paper is essentially minibatching a KL term for the $q(f|u)$. How does this work exactly and how do you scale all the terms.
- It’s unclear how the hyperparameters interact with the new approximation. Do they lead to better-calibrated predictive variances? Providing more details or empirical results on this aspect would help clarify the impact of the proposed method.
- In the non-Gaussian case, the paper resorts to the standard SVGP prediction, as using the proposed approach directly would be computationally too expensive. This suggests that the main benefit of the method in this setting is primarily improved hyperparameter estimation rather than a fundamentally better approximation of the posterior. If so you should also mention some related work in this ara.

**Relation To Broader Scientific Literature:**

Yes, the key contributions of the paper are well-situated within the broader scientific literature.

**Theoretical Claims:**

I went over the proofs, and they appear to be correct. The derivations follow standard results from the literature, and the steps are well-structured and logically sound.

---

> ### Author Rebuttal · Authors · 2025-04-01
>
> We would like to thank the reviewer. We respond to the main comments below.
>
> > Yes, mainly. The proposed methods and evaluation criteria are appropriate for the problem at hand, and the experiments effectively demonstrate the benefits of the tighter variational bound. However, the datasets used in the evaluation seem somewhat simple, and the paper could have further strengthened its empirical validation by including a more complex benchmark, such as MNIST or a real-world dataset. This would provide additional evidence of the method’s scalability and effectiveness in practical applications.
>
> Thank you for this comment. Here, we have focused mainly on the GP regression, large GP regression and we provide
> also an experiment with Poisson regression which is a non-Gaussian likelihood example.
>
> > The main weakness of the paper is that the results demonstrate only marginal improvements with the new bound across all evaluated tasks, without a clear example where the standard approach would fail without it. This makes it difficult to assess the practical necessity of the proposed refinement. Additionally, in the non-Gaussian likelihood setting, the paper opts for a simpler approximation rather than leveraging the more expressive variational form, seemingly due to computational constraints.
>
> Please note that we do observe noticeable improvements in predictive performance in some experiments (Pol, Bike, Kin40k, Protein, Buzz). For some datasets, like Pol and Kin40k, the improvement is significant. We believe that a very practical feature of the new ELBO is that in GP regression it requires a minor modification to existing code. So a practitioner can easily train a GP with the new ELBO
> while keeping the computational cost the same as in the previous SVGP bound.
>
> > Tighter sparse variational Gaussian processes, Bui et al (2025), Under review TMLR.
>
> Thank you for pointing to concurrent work. We will cite this work in the next version of our paper.
>
> > Table 2 is confusing. The top methods are baselines but have nothing to do with your method I assume. The bottom methods both have ours but isn't your method only SVGP-new?
>
> We agree with the reviewer.  We will modify the table to keep "ours" only for SVGP-new as suggested by the reviewer. The upper part of table gives two strong baselines from the literature (discussed in the first paragraph in the Related Work) that are based on different-type of extensions of the SVGP that allow to increase the number of inducing points. Indeed, they are unrelated to our method which replaces
> $p(f|u)$ by $q(f|u)$, but we have included them for comparison reasons.
>
> > When moving to the minibatch setting, the paper is essentially minibatching a KL term for the $q(f|u)$
> . How does this work exactly and how do you scale all the terms.
>
> Starting from the following expression for the ELBO (Equation 18 in the paper)
> $$
> \sum_{i=1}^N \left(  E_{q(u)} [\log \mathcal{N}(y_i | k_{f_i u} K_{u u}^{-1} u, \sigma^2 )]  - \frac{1}{2}  \log (1+\frac{k_{ii} - q_{ii}}{\sigma^2})  \right)   - \text{KL}[q(u) || p(u)],
> $$
> we observe that each term in the sum (i.e., the full term inside the big brackets) depends only on a single data point $(x_i, y_i)$, which is what is needed for minibatch training. Based on this, we can obtain an unbiased ELBO (and gradient) using a minibatch of $b$ data as
>  $$
> \frac{N}{b} \sum_{i \in minibatch} \left(  E_{q(u)} [\log \mathcal{N}(y_i | k_{f_i u} K_{u u}^{-1} u, \sigma^2 )]  - \frac{1}{2}  \log (1+\frac{k_{ii} - q_{ii}}{\sigma^2})  \right)  - \text{KL}[q(u) || p(u)],
> $$
> where note also that $E_{q(u)} [\log \mathcal{N}(y_i | k_{f_i u} K_{u u}^{-1} u, \sigma^2 )]$ is analytic. We can include an appendix to describe the above details.
>
> > It’s unclear how the hyperparameters interact with the new approximation. Do they lead to better-calibrated predictive variances? Providing more details or empirical results on this aspect would help clarify the impact of the proposed method.
>
> After training with the new ELBO, we still use the previous standard SVGP predictive density as discussed in Section 3.1.
> So the only difference is that the new ELBO can provide different hyperparameters (and inducing inputs $Z$).  Figure 1 and Table 1, 2 indicate that this can lead to better predictions in terms of test log-likelihoods. We can try to add an ablation to further study
> the effect on the predictive variances.
>
> | In the non-Gaussian case, the paper resorts to the standard SVGP prediction, as using the proposed approach directly would be computationally too expensive. This suggests that the main benefit of the method in this setting is primarily improved hyperparameter estimation rather than a fundamentally better approximation of the posterior. If so you should also mention some related work in this ara.
>
> Just to clarify here that for both Gaussian and non-Gaussian likelihoods we do predictions with previous SVGP predictive
> equations. We can clarify this further in the paper.

---

### Official Review · Reviewer_Rdie · 2025-03-11

**Overall Recommendation:** 4

**Summary:**

The paper introduces new evidence lower bounds (ELBOs) for sparse variational Gaussian processes (SVGP) by relaxing the traditional assumption that the variational distribution must factorize with the conditional GP prior p(f|u). Instead, the authors propose a more flexible variational distribution q(f|u), which allows for a tighter bound. Theoretical analysis shows that the new bound is provably tighter than previous SVGP bounds, and experiments on regression and non-Gaussian likelihood tasks demonstrate improved hyperparameter learning and predictive performance. However, the proposed methods between theoretical analysis and practical experiments are slightly different, which reduces the soundness of this paper. The practical method is computationally efficient, requires minimal code modifications, and is compatible with stochastic optimization.

**Claims And Evidence:**

The claims are supported by clear theoretical derivations (Lemmas 3.1–3.3, Proposition 3.5) and extensive experiments. The theoretical proofs are straightforward, and experiments on synthetic/real-world datasets (e.g., Snelson, UCI datasets, NYBikes) validate reduced bias in hyperparameters (e.g., noise variance) and better test log-likelihoods.

**Essential References Not Discussed:**

NA

**Experimental Designs Or Analyses:**

The comprehensive experiments cover regression (Gaussian/non-Gaussian) and varying dataset sizes.

**Methods And Evaluation Criteria:**

The methods are appropriate for scalable GP inference. The evaluation uses standard metrics (test log-likelihood, RMSE) and datasets (UCI, Kin40k), with comparisons to common baselines (SGPR, SVGP, SOLVE-GP). Experiments include multiple trials to report standard errors, ensuring statistical reliability.

**Other Comments Or Suggestions:**

Typos: "hyperpaparameters" (Abstract), "minibathes" (Section 3.2).

**Other Strengths And Weaknesses:**

NA

**Questions For Authors:**

The insight presented in this paper is promising, but the details may require further discussion. The key question is: to what extent does assuming a spherical V influence the results? The potential risks associated with this assumption are not explored, nor is there a theoretical or experimental analysis addressing its impact.

**Relation To Broader Scientific Literature:**

The work builds on SVGP, positioning the new bound as a tighter alternative. This applies to all of the sparse GP utilizing the inducing points.

**Theoretical Claims:**

The key insight, replacing p(f|u) with a diagonal-covariance q(f|u), is sound. However, the gap between diagonal V and spherical V is not thoroughly assessed.

---

> ### Author Rebuttal · Authors · 2025-04-01
>
> Thank you for your comments. Below we provide some responses.
>
> > The key insight, replacing $p(f|u)$ with a diagonal-covariance $q(f|u)$, is sound. However, the gap between diagonal V and spherical V is not thoroughly assessed.
>
> Please note that in the medium-size regression experiments reported in Table 1 and Figure 2 we do compare with the method that assumes spherical $V = v^* I$ which uses the optimal scalar value $v^* = \left( 1 + \frac{\text{tr}(K_{f f} - Q_{f f})}{N \sigma^2} \right)^{-1}$. This is precisely, the method denoted in the experiments as "SGPR-artemev".  We will clarify this further in the paper. Note, that currently we briefly explain the spherical case $V = v I$  for GP regression and the connection with Artemev et al's bound in Related Work (Section 4) and also in Appendix B.4.  From Table 1 and Figure 2 we can observe that diagonal V does work better than the scalar $v$.
>
> > The insight presented in this paper is promising, but the details may require further discussion. The key question is: to what extent does assuming a spherical V influence the results? The potential risks associated with this assumption are not explored, nor is there a theoretical or experimental analysis addressing its impact.
>
> We agree with the reviewer that the fact that the non-Gaussian likelihood case requires spherical $V = v I$ is not ideal.  In seems that for such non-Gaussian likelihoods the only option that works is to use a spherical $V = v I$ and, as explained in Section 3.3, the diagonal $V$ (that works for GP regression) has cubic cost when trying to obtain each marginal $q(f_i)$. Notice also that if someone heuristically tries to use diagonal $V$ for the non-Gaussian likelihood case and approximate the marginal $q(f_i)$ by
> $$
> q(f_i)  = \mathcal{N}(f_i | k_{f_i u} K_{u u}^{-1} m, v_i (k_{ii} - q_{ii}) + k_{f_i u} K_{u u}^{-1} S K_{u u}^{-1} k_{u f_i}),
> $$
> (which is not the correct marginal under $q(f|u) q(u)$ since the variance term  $v_i (k_{ii} - q_{ii})$  is wrong) then this creates an inconsistency in the variational distribution in the ELBO, since the $q(f_i)$ used to compute the expected log-likelihood term will be inconsistent with the $q(f|u) q(u)$ in the KL divergence term $KL[q(f|u) q(u) || p(f|u) p(u)]$), and the objective is not a rigorous ELBO anymore.
>
> We can add further discussion to clarify that the spherical $V= v I$ is needed for non-Gaussian likelihoods in order to obtain a rigorous ELBO.

---

### Official Review · Reviewer_kDa9 · 2025-03-12

**Overall Recommendation:** 5

**Summary:**

The authors present an improvement on the standard SVGP approximation by departing from the standard conditional GP prior distribution. The approach introduces an additional $N$ variational parameters which modify the covariance matrix of the conditional distribution. This leads to an improvement on the resultant bound for the log marginal likelihood. The authors demonstrate that their method works in the stochastic minibatch optimisation setting, and can be extended effectively to non-Gaussian likelihoods by judicious constraints on the additional variational parameters. The theoretical findings are shown to hold in practice on a number of small to large scale regression experiments.

**Claims And Evidence:**

Yes---claims are theoretical and demonstrated through experiments.

**Essential References Not Discussed:**

N/A.

**Experimental Designs Or Analyses:**

Yes.

**Methods And Evaluation Criteria:**

Yes.

**Other Comments Or Suggestions:**

N/A.

**Other Strengths And Weaknesses:**

## Strengths

* The proposed approach is novel (together with https://arxiv.org/abs/2502.04750 which, coincidentally, was released at the same time), straightforward to both understand and implement, and effective in practice, demonstrating improved or on-par performance relative to baselines across almost all experiments. Collectively, the paper is very convincing.


## Weaknesses

* A key strength of the proposed method is in its generality---I suspect that it can be applied to improve a wealth of SVGP approximations such as SOLVE-GP, and different forms of approximations such those used for SVGP-LVMs and Deep-SVGPs. I believe that further extensions in the paper would improve it further, although the authors do touch upon this as future work.

**Questions For Authors:**

N/A.

**Relation To Broader Scientific Literature:**

N/A.

**Theoretical Claims:**

Yes.

---

> ### Author Rebuttal · Authors · 2025-04-01
>
> Thank you for very accurately describing the contribution of the paper and for pointing to concurrent work. As also mentioned in the response to Reviewer Pyzb below, we plan to discuss the concurrent work in the next version of our paper.

---

### Decision · Program_Chairs · 2025-05-01

**Decision:**

Accept (spotlight poster)

**Comment:**

The sparse variational approach of Titsias (2009) and its siblings has enabled computationally tractable inference and training of Gaussian process models, allowing stochastic data mini-batching, non-Gaussian likelihoods, and efficient inference in hierarchical and latent variable settings. One key assumption in this approach is that the factor p(f|u) is retained in the variational approximation, q(f|u) = p(f|u). This submission proposes a new structured q(f|u) that results in tighter variational bounds whilst maintaining tractability. The proposal is an interesting contribution to the sparse GP literature, simple to implement and yields marginal but consistent practical gains. The reviewers were very positive and in agreement that this is a clear accept.